# Diurnal or Nocturnal? Federated Learning of Multi-branch Networks from Periodically Shifting Distributions

**Chen Zhu**[1]*, **Zheng Xu**[2], **Mingqing Chen**[2], **Jakub Konecný**[2], **Andrew Hard**[2], **Tom Goldstein**[1]
[1] University of Maryland, College Park, [2] Google

## Abstract

Federated learning has been deployed to train machine learning models from decentralized client data on mobile devices in practice. The clients available for training are observed to have periodically shifting distributions changing with the time of day, which can cause instability in training and degrade the model performance. In this paper, instead of modeling the distribution shift with a block-cyclic pattern as previous works, we model it with a mixture of distributions that gradually shifts between daytime and nighttime modes, and find this intuitive model to better match the observations in practical federated learning systems. Furthermore, we propose to jointly train a clustering model and a multi-branch network to allocate lightweight specialized branches to clients from different modes. A temporal prior is used to significantly boost the training performance. Experiments for image classification on EMNIST and CIFAR datasets, and next word prediction on the Stack Overflow dataset show that the proposed algorithm can counter the effects of the distribution shift and significantly improve the final model performance.

## 1 Introduction

In Federated Learning (FL), many clients collaboratively train a machine learning model with decentralized data under the orchestration of a central server (Kairouz et al., 2019). FL is designed for privacy protection: the private data of local clients will never be directly transferred to the server or shared with other clients, which follows the principle of data minimization and keeps the attack surface of the system small (Wang et al., 2021). Initially introduced for decentralized training on mobile devices (McMahan et al., 2017), FL has been widely applied for various different applications including finance, health, digital assistance and personalized recommendations (see a few recent surveys (Yang et al., 2019; Kairouz et al., 2019; Li et al., 2020a; Lim et al., 2020; Wang et al., 2021)). Specifically, cross-device FL has been used in practice to improve utility and privacy of applications such as next word prediction (Hard et al., 2018), emoji suggestion (Ramaswamy et al., 2019), query suggestion (Yang et al., 2018), out-of-vocabulary word discovery (Chen et al., 2019), and keyword trigger models (Granqvist et al., 2020; Hard et al., 2020).

A typical communication round of FL starts with a server broadcasting a global model to clients. Clients then perform local computation on private data and only send back aggregated model updates. Finally, the server aggregates the client updates and apply them to the global model before beginning the next round. In practical FL systems (Bonawitz et al., 2019; Paulik et al., 2021), clients can only participate when the local criteria is met, such as when mobile devices are charging, idle, and connected to an unmetered network. For the server, clients that satisfy their local criteria and participate training at different times of the day are usually from different time zones that can have significant differences, which can cause a periodically shifting data distribution that may degrade the training stability and final model performance (Yang et al., 2018; Eichner et al., 2019). For centralized systems where client data can be collected, such a problem may be mitigated by caching and uniformly sampling from cached data. However, due to the privacy and system constraints (Wang

---

*Work done as an intern at Google. Correspondence to: `chenzhu@umd.edu, xuzheng@google.com`.

et al., 2021), the orchestrator (server) in FL systems is not allowed to collect the raw user data, and must deal with such non-IID, heterogeneous data distribution.

To our knowledge, there are only a few previous works (Eichner et al., 2019; Ding et al., 2020) discussing periodical distribution shift of client population in federated learning. These works assume a block-cyclic structure where daytime clients and nighttime clients alternately participate in training. Eichner et al. (2019) proposed the semi-cyclic SGD approach, where clients participating training at different time slots will contribute to and only use the corresponding model of the group. By learning separate models for different blocks, they can obtain the same guarantee as the i.i.d. non-cyclic setting under their assumptions. However, there are several caveats of semi-cyclic SGD that makes it difficult to apply in practice: (1) It assigns models to clients based on their participation time, but not all clients will participate in federated learning, hence it is hard to decide the correct group for these clients. (2) It maintains a version of the full model for each clients group, which potentially increases the communication cost or privacy risk. (3) The assumption of the abrupt switch from the daytime group to the nighttime group at a specific time of a day is unintuitive in practice. (Ding et al., 2020) is a variant of semi-cyclic SGD that inherits these issues. We provide discussions of more related works such as heterogeneity, clustering, and multi-branch networks in Appendix A.7.

In this paper, we study periodical distribution shift of clients, and make the following contributions:

1. We revisit the periodical distribution shift. Instead of adopting the block-cyclic structure (Eichner et al., 2019), we assume a smooth transition between the daytime mode and nighttime mode, and empirically verify through simulation that its impact on training better matches the observation in practical FL systems.

2. We propose to jointly train a multi-branch network and a clustering model to select the branch that better fits the client's distribution based on the feature representations. The lightweight branches for the day and night modes only slightly increase the communication cost, but significantly improve the model performance. Unlike (Mansour et al., 2020; Ghosh et al., 2020; Marfoq et al., 2021), the feature-based clustering model does not rely on labelled data, and can be easily applied for inference on new clients.

3. We propose to use the temporal prior of the client distribution to enhance the clustering models. We assume participating clients per communication round is a mixture of daytime and nighttime clients, and the number of participating clients from the daytime group will gradually increase as time goes from nighttime to daytime, and vice versa. By exploiting this prior, we can train models that are even more accurate than models trained with uniformly sampled clients in a conventional federated simulation.

4. We provide simulations of the distribution shift on three benchmark datasets (EMNIST, CIFAR and Stack Overflow) to evaluate the empirical performance of FL algorithms under the periodic distribution shift with smooth transition. We perform extensive experiments, where the multi-branch networks trained by our method outperform the distribution-oblivious baselines by a large margin: 3-5% on EMNIST, 2-14% on CIFAR, and 0.4-1.35% on the challenging Stack Overflow dataset under various degrees of distribution shifts. By leveraging the temporal priors, the proposed method can take advantage of the periodic distribution shift and beat the strong baselines of training with uniformly sampled clients by 4%, 4% and 0.45%, respectively.

## 2 MODELING PERIODICAL DISTRIBUTION SHIFT

**FL setting.** We consider federated learning algorithms for a set of clients $\mathcal{I}$, where the $i$-th client has data $\mathcal{D}_i$ sampled IID (independent and identically distributed) from its own distribution, but the distribution of different clients can be heterogeneous. We minimize the expected loss on all clients,

$$\text{minimize}_{\boldsymbol{w}} \; L(\boldsymbol{w}) = \sum_{i \in \mathcal{I}} p_i L_i(\boldsymbol{w}), \text{ where } L_i(\boldsymbol{w}) = \frac{1}{|\mathcal{D}_i|} \sum_{\boldsymbol{\xi} \in \mathcal{D}_i} \ell(\boldsymbol{w}, \boldsymbol{\xi}), \text{ and } \sum_{i \in \mathcal{I}} p_i = 1, \quad (1)$$

where $p_i$ is the weight (probability) of client $i$, and $\boldsymbol{\xi} = (\boldsymbol{x}, \boldsymbol{y})$ is a training sample pair of data and label on a client. By setting $p_i = |\mathcal{D}_i| / \sum_{j \in \mathcal{I}} |\mathcal{D}_j|$, the federated training loss recovers the empirical risk minimization (ERM) objective on all client samples. For simplicity, we abuse notation and use $\boldsymbol{x} \in \mathcal{D}_i$ to denote a training sample (without label) for client $i$.

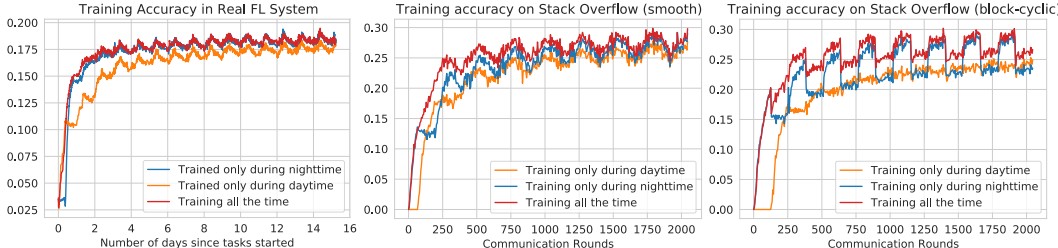

Figure 1: Training accuracy of language models. (Left): on-device training in a practical FL system. (Middle): simulation of the smooth distribution shift with $q_{L,1}(t)$ on Stack Overflow ($T = 256$). In the beginning of each period, the probability of clients coming from nighttime mode is 1, which linearly decreases to 0 at the middle of each period so that all clients are from daytime mode, corresponding to the peaks and valleys of the curves respectively. (Right): simulation of the block-cyclic shift on Stack Overflow ($T = 256$).

**Periodic distribution shift.** A subset of clients $\mathcal{I}'(t) \subset \mathcal{I}(t)$ are available for training in a communication round $t$. $\mathcal{I}(t)$ periodically changes with most of the clients from a daytime (nighttime) client group in midday (midnight). Figure 1 (left) shows that the training loss in a cross-device FL system has daily oscillation. Such oscillations was also observed by Yang et al. (2018), who conjecture it is due to the domain differences between clients from different time zones. Eichner et al. (2019) study the block-cyclic structure, where the model is trained for $T$ rounds each day, and the clients are from the day mode and night mode alternately, each last continuously for $T/2$ rounds. We plot the training curves of block-cyclic structure in Figure 1 (right), and observe it is different from the (left) curves from a practical FL systems.

**Smooth transition.** We also assume the distribution changes periodically with a period of $T$. Unlike (Eichner et al., 2019), we assume clients at round $t$ are a mixture of daytime clients and nighttime clients, denoted as $\mathcal{I}_1(t)$ and $\mathcal{I}_2(t)$, respectively. Intuitively, since the available population is usually large around the clock, the population distribution of available clients should shift gradually, rather than abruptly from one mode to another as in the block-cyclic structure. To better approximate the behavior in practice, we assume that in each period, clients come from the day mode $\mathcal{I}_1(t)$ with a probability $q(t)$ that varies *smoothly* between 0 and 1. Specifically, we simulate $\mathcal{I}_1(t)$ and $\mathcal{I}_2(t)$ with two disjoint sets of clients with different data distributions, and define $q : \mathbb{R}^+ \to [0, 1]$ to be a periodic function with a period of $T$. At each round $t$, we sample the clients from the following distribution

$$P(i \in \mathcal{I}_1(t)) = q(t), \ P(i \in \mathcal{I}_2(t)) = 1 - q(t). \tag{2}$$

We consider a periodic *Linear* function (L) and a smooth *Cosine* function (C) for $q(t)$, each further parameterized by an exponent factor $p > 0$ to control smoothness of the transition,

$$q_{L,p}(t) = \left| 2\frac{t \bmod T}{T} - 1 \right|^p, q_{C,p}(t) = \left[ \frac{1}{2}\left( \cos\left(2\pi t/T\right) + 1 \right) \right]^p. \tag{3}$$

We visualize the transition probability $q(t)$ in Figure 7 of the appendix. When $p < 1$, more daytime clients are available in $T$ rounds, and when $p > 1$, more nighttime clients are available. This can simulate the difference in the number of completed training rounds during daytime and nighttime observed in practice (Yang et al., 2018).

**Observation and insight.** Figure 1 (middle) simulates the training curve with $q_{L,1}(t)$ to control the probability for sampling from $\mathcal{I}_1(t)$, which more accurately approximates the curves from a practical FL system. The three curves show that training a single model around the clock achieves the best performance on both modes despite the domain differences, which motivates us to train a model with a large amount of shared weights. Semi-cyclic SGD (Eichner et al., 2019) provides the worst-case guarantees when the domains of each block are unrelated, and argues that training a single model on both modes is not optimal, which is different from our observations. Another important observation is that the training accuracy reaches its minima (maxima) when the client population is most biased towards the day mode or night mode, e.g., round 1024 and 1152 in the middle figure, from which we can infer the peak moment of daytime clients and nighttime clients in practice and use it to define a strong prior to improve the learning process.

## 3 LEARNING FROM PERIODICAL DISTRIBUTION SHIFT

We consider applications in which clients collaboratively train a model with a common input and output space, and the distribution of clients changes with time. We propose to jointly train a multi-branch network and a clustering model to assign specialized branches for prediction on clients from different modes, guided by the temporal prior of the client distributions. We consider two methods for enforcing the temporal prior: FEDTEM (Section 3.1), which is based on a Gaussian Mixture Model (GMM), and FEDTKM ( Section 3.2), which is based on K-means clustering.

**Multi-branch network.** Eichner et al. (2019) showed the merits of training separate models for different data distributions when the distributions can be identified during evaluation. However, training a single model with shared feature extraction layers on all available data can usually improve the data efficiency of representation learning. For example, for vision tasks, it helps to learn to extract common low-level features, while for language tasks it helps to learn shared embeddings and grammars from the context. To handle the distribution shift while learning shared feature representations and alleviating the communication overhead, we adopt the weight-sharing strategy from multi-task learning (Caruana, 1997) to train a multi-branch network with shared feature extraction layers $f(\boldsymbol{w}_f, \boldsymbol{x})$ followed by one of the specialized prediction branches $g_k(\boldsymbol{w}_k, f(\boldsymbol{w}_f, \boldsymbol{x}))$ for clients from each cluster $k$ ($1 \leq k \leq K$). We set each prediction branch $g_k$ to be a single linear layer, which is more communication efficient and data efficient than having $K$ versions of the same model.

**Temporal prior.** The temporal prior $\tilde{q}(t)$ is an estimate for the ratio of clients coming from the daytime cluster, $q(t)$. From the observations in Section 2, we can locate the time when $q(t)$ is most likely to be 0 or 1 by observing when the minima and maxima occur from the training curve. In between these minima and maxima, we consider three types of $\tilde{q}(t)$ in our current experiments: 1) Linear: $\tilde{q}(t) = q_{L,1}(t)$; 2) Cosine: $\tilde{q}(t) = q_{C,1}(t)$; 3) Soft: see Appendix A.3.

**Training Objective.** *Our model assumes clients are from either the daytime or the nighttime cluster* ($K = 2$), so it has two branches, and each branch is used for one group of clients. Let $k_i^*$ be the branch index of client $i$ chosen by the clustering model, and $\bar{k}_i^*$ be the index of the other branch. Each client trains the network with the following objective

$$\underset{\boldsymbol{w}}{\text{minimize}}\ L_i(\boldsymbol{w}) = \frac{1}{|\mathcal{D}_i|} \sum_{\boldsymbol{\xi} \in \mathcal{D}_i} \ell_{CE}(g_{k_i^*}(\boldsymbol{w}, \boldsymbol{x}), \boldsymbol{y}) + \lambda \ell_{CE}(g_{\bar{k}_i^*}(\boldsymbol{w}, \boldsymbol{x}), s(\epsilon, \boldsymbol{y})), \quad (4)$$

where $s(\epsilon, \boldsymbol{y}) = \epsilon \frac{1}{n} + (1 - \epsilon)\boldsymbol{y}$ is the label smoothing function for one-hot vector $\boldsymbol{y}$, $\epsilon \in [0, 1]$ determines the amount of label smoothing, and $\lambda > 0$ is the regularization strength. The label smoothing regularization updates the other branch jointly with the feature extractor to prevent staleness, while encouraging the two branches to specialize in different feature subspaces: branch $\bar{k}_i^*$ is trained to become less certain on features of samples from cluster $k_i^*$.

### 3.1 LEARNING A GAUSSIAN MIXTURE MODEL WITH PER-CLIENT TEMPORAL STATISTICS

We propose Federated Expectation-Maximization with Temporal prior (FEDTEM) to learn a Gaussian Mixture Model (GMM) to infer the cluster a client is from, and select the corresponding branch in the multi-branch network. We define discrete latent variables $z$ and $\zeta$ to represent which cluster a sample and a client is from, respectively. FEDTEM assumes samples on the same client are from the same cluster, so the GMM prior $P(\zeta = k) = P(z = k)$ for any $k$. We define $P(\boldsymbol{x}|z = k)$ as a Gaussian distribution $\mathcal{N}(f(\boldsymbol{w}, \boldsymbol{x})|\boldsymbol{\mu}_k, \boldsymbol{\sigma}_k)$ in the feature space. For efficiency, we assume and learn a diagonal covariance $\boldsymbol{\sigma}_k$ for the Gaussian models. Algorithm 1 summarizes the training process, and the details of each step are provided in the following sections.

### 3.1.1 MODELING THE CLIENT DISTRIBUTIONS AND SELECTING BRANCHES FOR TRAINING

The prior $P(\zeta)$ of the client distribution are constantly changing due to the periodic distribution shift. To stay up-to-date, the $i$-th client estimates its probability of coming from the $k$-th cluster based on its data $\mathcal{D}_i$ before the local update steps during training. Specifically, given the GMM parameters, since $P(\zeta) = P(z)$, the local maximum likelihood estimation (MLE) of $P(\zeta_i)$ on each new client $i$ is equal to the MLE of $p(z)$ on client $i$. Let $\boldsymbol{\pi}_i^*$ be the MLE of $P(z)$ on client $i$, where

the $k$-th dimension of $\boldsymbol{\pi}_i^*$, denoted as $\pi_{ik}^*$, is the MLE of $P(z = k)$ on client $i$. Then

$$P(\zeta_i = k) = \pi_{ik}^* = \frac{1}{|\mathcal{D}_i|} \sum_{\boldsymbol{x} \in \mathcal{D}_i} P(z = k|\boldsymbol{x}) = \frac{1}{|\mathcal{D}_i|} \sum_{\boldsymbol{x} \in \mathcal{D}_i} \frac{\pi_k \mathcal{N}(f(\boldsymbol{w}, \boldsymbol{x})|\boldsymbol{\mu}_k, \boldsymbol{\sigma}_k)}{\sum_{j=1}^{K} \pi_j \mathcal{N}(f(\boldsymbol{w}, \boldsymbol{x})|\boldsymbol{\mu}_j, \boldsymbol{\sigma}_j)}, \quad (5)$$

For completeness, we give the derivation of Eq. 5 in Appendix A.1, where we also compare with the empirical results of using the posterior $P(\zeta|\mathcal{D}_i)$ instead of the MLE $\boldsymbol{\pi}_i^*$ for branch selection.

We select the branch based on the MLE of $P(\zeta_i)$ and train the network by optimizing Eq. 4. We can either greedily select the branch $k_i^* = \arg\max_k \boldsymbol{\pi}_i^*$, or sample from the discrete distribution $\boldsymbol{\pi}_i^*$. We show the greedy approach achieves better empirical results in Figure 6, and use it by default.

---

**Algorithm 1** FEDTEM: Federated EM with Temporal Prior (Training)

---

1: **Input:** A stream of clients $\mathcal{I}(t)$ with periodical distribution shift; Number of communication rounds $N$; Number of rounds per day $T$.
2: **Output:** Network parameters $\boldsymbol{w}^N = (\boldsymbol{w}_f^N, \boldsymbol{w}_1^N, \boldsymbol{w}_2^N)$, GMM parameters $\boldsymbol{\theta}^N = (\boldsymbol{\mu}^N, \boldsymbol{\sigma}^N, \boldsymbol{\pi}^N)$.
3: **for** $t = 0$ to $N - 1$ **do**
4:     A set of $m$ clients $\mathcal{I}'(t) \subset \mathcal{I}(t)$ participates training;
5:     Server broadcasts parameters of the network $\boldsymbol{w}^t$ and the GMM $\boldsymbol{\theta}^t$ to $\mathcal{I}'(t)$;
6:     **for** clients $i \in \mathcal{I}'(t)$ **in parallel do**
7:         Estimate $\boldsymbol{\pi}_i^*$ on $\mathcal{D}_i$, and choose the branch $k_i^* = \arg\max_k \pi_{ik}^*$;            ▷ see Eq. 5
8:         Given $k_i^*$, run local updates for the network by optimizing Eq. 4 and get $\tilde{\boldsymbol{w}}_i^{t+1}$;
9:         On $\mathcal{D}_i$, compute the MLE $\tilde{\boldsymbol{\pi}}_i^*$ and all feasible Gaussian parameters $(\tilde{\boldsymbol{\mu}}_i^{t+1}, \tilde{\boldsymbol{\sigma}}_i^{t+1})$;      ▷ see Eq.9
10:     Server aggregates the model update $\sum p_i(\tilde{\boldsymbol{w}}_i^{t+1} - \boldsymbol{w}^t)$
11:     Update network parameters to $\boldsymbol{w}^{t+1}$ with the prescribed server optimizer.
12:     Server collects GMM params $\{\tilde{\boldsymbol{\pi}}_i^*, \tilde{\boldsymbol{\mu}}_i^{t+1}, \tilde{\boldsymbol{\sigma}}_i^{t+1} | i \in \mathcal{I}'(t)\}$;
13:     Update the GMM parameters to $(\boldsymbol{\mu}^{t+1}, \boldsymbol{\sigma}^{t+1}, \boldsymbol{\pi}^{t+1})$ with the temporal prior on $\bar{\boldsymbol{\pi}}_i^*$;     ▷ see Eq. 10

---

### 3.1.2 UPDATING THE MIXTURE MODEL PARAMETERS

We introduce a federated Expectation-Maximization (EM) (Dempster et al., 1977) enhanced by the temporal prior to update the parameters of the GMM. The temporal prior is enforced via a bottom-up approach: we start by running EM on each client $i$ and send all possible locally optimal GMM parameters to the server, and then select and aggregate the GMM parameters with the temporal prior on the server. We find the optimal GMM parameters after the local update steps as described in Section 3.1.1, so that the GMM is updated in the feature space of the updated network. We give details of the process in the following.

**E step.** For each client $i \in \mathcal{I}(t)$, evaluate the posterior $\gamma_{ik}(\boldsymbol{x})$ for each sample $\boldsymbol{x}$ to infer the probability of $\boldsymbol{x}$ coming from cluster $k$, based on the locally updated network with parameters $\boldsymbol{w}_i^{t+1}$

$$\gamma_{ik}(\boldsymbol{x}) = P(z = k|\boldsymbol{x}) = \frac{\pi_k^t \mathcal{N}(f(\boldsymbol{w}_i^{t+1}, \boldsymbol{x})|\boldsymbol{\mu}_k^t, \boldsymbol{\sigma}_k^t)}{\sum_{j=1}^{K} \pi_j^t \mathcal{N}(f(\boldsymbol{w}_i^{t+1}, \boldsymbol{x})|\boldsymbol{\mu}_j^t, \boldsymbol{\sigma}_j^t)}. \quad (6)$$

**M step on clients.** To learn a GMM that better distinguishes the daytime and nighttime clusters, we incorporate the temporal prior $\tilde{q}(t)$ into the M step by considering the posterior $P(\zeta = k|\mathcal{D}_i, \boldsymbol{\theta}^t, \tilde{q}(t))$, where $\boldsymbol{\theta}^t = (\boldsymbol{\mu}^t, \boldsymbol{\sigma}^t, \boldsymbol{\pi}^t)$ is the collection of all GMM parameters. The M step optimizes the following objective

$$\underset{\boldsymbol{\theta}}{\text{maximize}} \sum_{i \in \mathcal{I}'(t)} \sum_{k=1}^{K} P(\zeta = k|\mathcal{D}_i, \boldsymbol{\theta}^t, \tilde{q}(t)) \log P(\mathcal{D}_i, \zeta = k|\boldsymbol{\theta}, \tilde{q}(t)). \quad (7)$$

Since the clients cannot share their data with the server, we first find locally optimal parameters $\tilde{\boldsymbol{\theta}}_i^{t+1} = \arg\max_{\boldsymbol{\theta}} \sum_{k=1}^{K} P(\zeta = k|\mathcal{D}_i, \boldsymbol{\theta}^t, \tilde{q}(t)) \log P(\mathcal{D}_i, \zeta = k|\boldsymbol{\theta}, \tilde{q}(t))$ on each client $i$, and then average on the server. However, it is hard to evaluate the posterior $P(\zeta = k|\mathcal{D}_i, \boldsymbol{\theta}^t, \tilde{q}(t))$ locally without additional communication rounds, since $\tilde{q}(t)$ is an estimate of the ratio of clients from one cluster, and the posterior for one client depends on the estimates of other clients. A viable approach is to assume the posterior to be one-hot, i.e., for a certain cluster $k^*$, $P(\zeta = k|\mathcal{D}_i, \boldsymbol{\theta}^t, \tilde{q}(t)) = 1$ if $k = k^*$, and $P(\zeta = k|\mathcal{D}_i, \boldsymbol{\theta}^t, \tilde{q}(t)) = 0$ if $k \neq k^*$. Under this modeling assumption, we can learn

more distinctive features for the clusters at no communication overhead. To see this, when $k^*$ is given, we only need to solve

$$\tilde{\boldsymbol{\theta}}_{ik^*}^{t+1} = \text{argmax}_{\boldsymbol{\theta}} P(\zeta = k^*|\mathcal{D}_i, \boldsymbol{\theta}^t, \tilde{q}(t)) \log P(\mathcal{D}_i, \zeta = k^*|\boldsymbol{\theta}, \tilde{q}(t)), \tag{8}$$

where its optimal mean and variance, and the MLE of the GMM prior for cluster $k^*$ on client $i$ are

$$\tilde{\boldsymbol{\mu}}_{ik^*}^{t+1} = \frac{1}{|D_i|} \sum_{\boldsymbol{x} \in \mathcal{D}_i} f(\boldsymbol{w}_i^{t+1}, \boldsymbol{x}), \ [\tilde{\boldsymbol{\sigma}}_{ik^*}^{t+1}]^2 = \frac{1}{|\mathcal{D}_i|} \sum_{\boldsymbol{x} \sim \mathcal{D}_i} (f(\boldsymbol{w}_i^{t+1}, \boldsymbol{x}) - \tilde{\boldsymbol{\mu}}_{ik^*}^{t+1})^2, \tilde{\pi}_{ik^*}^* = \frac{1}{|\mathcal{D}_i|} \sum_{\boldsymbol{x} \in \mathcal{D}_i} \gamma_{ik^*}(\boldsymbol{x}). \tag{9}$$

Note $\tilde{\boldsymbol{\mu}}_{ik^*}^{t+1}$ and $\tilde{\boldsymbol{\sigma}}_{ik^*}^{t+1}$ are identical for different $k^*$; see Appendix A.2 for derivations. The clients send the optimal mean and variance to the server. To enforce temporal priors on the server, clients also send the MLE of the prior on $\mathcal{D}_i$ for each cluster $k$, i.e., $\tilde{\pi}_{ik}^*$, to the server.[1] In this way, the total number of parameters sent to the server can be less than a single GMM model.

**M step on server with temporal prior.** The temporal prior is enforced via a bottom-up approach: each client sends all possible solutions and MLEs $\{(\tilde{\boldsymbol{\mu}}_{ik}^{t+1}, \tilde{\boldsymbol{\sigma}}_{ik}^{t+1}, \tilde{\pi}_{ik}^*)|i \in \mathcal{I}'(t), k \in \{1, 2\}\}$ to the server; the server then estimates the one-hot posterior $P(\zeta|\mathcal{D}_i, \boldsymbol{\theta}^t, \tilde{q}(t))$ based on $\tilde{\pi}_{ik}^*$. A set $\tilde{\mathcal{I}}_1'(t)$ of $\lfloor \tilde{q}(t) \cdot |\mathcal{I}'(t)| + \frac{1}{2} \rfloor$ clients with highest $\tilde{\pi}_{i1}^*$ will have $P(\zeta = 1|\mathcal{D}_i, \boldsymbol{\theta}^t, \tilde{q}(t)) = 1$, while the remaining clients, constituting a set $\tilde{\mathcal{I}}_2'(t)$, will have $P(\zeta = 2|\mathcal{D}_i, \boldsymbol{\theta}^t, \tilde{q}(t)) = 1$. Then, similar as FedAvg (McMahan et al., 2017), the server updates the GMM parameters for each mode $k$ as the weighted average of the optimal GMM parameters of each client $i \in \tilde{\mathcal{I}}_k'(t)$:

$$\boldsymbol{\mu}_k^{t+1} = \sum_{i \in \tilde{\mathcal{I}}_k'(t)} \frac{|\mathcal{D}_i|}{M_k^t} \tilde{\boldsymbol{\mu}}_{ik}^{t+1}, \ [\boldsymbol{\sigma}_k^{t+1}]^2 = \sum_{i \in \tilde{\mathcal{I}}_k'(t)} \frac{|\mathcal{D}_i|}{M_k^t} [\tilde{\boldsymbol{\sigma}}_{ik}^{t+1}]^2, \ \pi_k^{t+1} = \frac{M_k^t}{\sum_{j=1}^K M_j^t}, \tag{10}$$

where $M_k^t = \sum_{i \in \mathcal{I}_k'(t)} |\mathcal{D}_i|$ is the total number of samples from clients assigned to mode $k$. Note $\boldsymbol{\sigma}^{t+1}$ is not necessarily an unbiased estimate, but we find this estimate gives good results in practice. We also use a running average of the GMM prior $\pi_k^{t+1}$ during training and set it to be a uniform distribution during inference; see Appendix A.4 for details.

## 3.2 Learning a Clustering Model with Aggregated Temporal Statistics

FEDTEM collects per-client $\tilde{\pi}_i^*$ to use the temporal prior and update the GMM parameters, which may not satisfy the strong aggregation-only data minimization principle (Bonawitz et al., 2021; Wang et al., 2021). We propose an alternative Federated $K$-Means algorithm augmented by the Temporal prior (FEDTKM), where the temporal prior is enforced based on only aggregated results, described in Algorithm 2. FEDTKM clusters the $i$th client and selects the branches based on the averaged distance of the features to the $k$ cluster centers,

$$d_{ik}(\boldsymbol{w}^t) = \frac{1}{|\mathcal{D}_i|} \sum_{\boldsymbol{x} \in \mathcal{D}_i} \alpha_k \| f(\boldsymbol{w}^t, \boldsymbol{x}) - \boldsymbol{c}_k \|, \tag{11}$$

where the superscript $t$ denotes the communication round, and $\alpha_k$ are distance scalar factors to control the "prior" of the clusters and enforce the temporal priors. During training, a client $i$ computes $d_{ik}$ on its training set to select the branch with minimal $d_{ik}$, while at test time, we use a minibatch to estimate $d_{ik}$. In the case of two centers, we fix $\alpha_1 = 1$, and learn a scalar $\alpha_2 > 0$ to rescale the distances to the *second* cluster $\boldsymbol{c}_2$: when $\alpha_2 > 1$, it makes the FEDTKM algorithm more likely to assign the client to the *first* cluster, and vice versa. $\alpha_2 > 1$ is updated by a quantile-based (private) estimator inspired by (Andrew et al., 2021).

The clients train the network with their selected branches by Eq. 11 through the objective in Eq. 4. After the local training steps, each client estimates its cluster assignment again with the locally updated parameters $\tilde{\boldsymbol{w}}_i^t$, and computes an indicator variable $q_i(t) = \mathbb{1}[d_{i1}(\tilde{\boldsymbol{w}}_i^t) < d_{i2}(\tilde{\boldsymbol{w}}_i^t)]$ for quantile estimation, and the feature means $\bar{f}(\tilde{\boldsymbol{w}}_i^t) = \frac{1}{|\mathcal{D}_i|} \sum_{\boldsymbol{x} \in \mathcal{D}_i} f(\tilde{\boldsymbol{w}}_i^t, \boldsymbol{x})$ for $k$-means centers. Then $q_i(t)$ and $\bar{f}(\tilde{\boldsymbol{w}}_i)$ can be sent to a trusted, secure aggregator to get the ratio of clients assigned to cluster 1, $\bar{q}(t)$, and the averaged $k$-means centers $\boldsymbol{c}_1^{t+1}, \boldsymbol{c}_2^{t+1}$ as

$$\bar{q}(t) = \frac{1}{|\mathcal{I}'(t)|} \sum_{i \in \mathcal{I}'(t)} q_i(t), \ \boldsymbol{c}_k^{t+1} = \sum_{i \in \mathcal{I}'(t)} \frac{r_{ik}|\mathcal{D}_i|}{\sum_{i \in \tilde{\mathcal{I}}'(t)} r_{ik}|\mathcal{D}_i|} \bar{f}(\tilde{\boldsymbol{w}}_i^t), \ k = 1 \text{ or } 2, \tag{12}$$

---

[1]The MLE is defined in Eq. 14.

where $r_{i1} = q_i(t), r_{i2} = 1 - q_i(t)$.

The distance factor $\alpha_2$ is updated by tracking the difference between the empirical quantile $\bar{q}(t)$ and the oracle quantile $\tilde{q}(t)$ estimated by the temporal prior,

$$\alpha_2^{t+1} = \exp\left\{\eta^t \left[\tilde{q}(t) - \bar{q}(t)\right]\right\} \alpha_2^t, \tag{13}$$

where $\eta^t \geq 0$ is the step size of the geometric update at step $t$. In all experiments, we use the periodical linear function $\tilde{q}(t) = q_{L,1}(t)$ as the temporal prior.

---

**Algorithm 2** FEDTKM: Federated $k$-means with Temporal Prior

---

1: **Input:** A stream of clients $\mathcal{I}(t)$ from a periodically shifting distribution; Number of communication rounds $N$; Number of rounds per day $T$.
2: **Output:** Network parameters $\boldsymbol{w}^N = (\boldsymbol{w}_f^N, \boldsymbol{w}_1^N, \boldsymbol{w}_2^N)$, $k$-means cluster centers $\boldsymbol{c}_1^N$ and $\boldsymbol{c}_2^N$ ($k = 2$), distance scalar $\alpha_2^N$.
3: **for** $t = 0$ **to** $N - 1$ **do**
4:     A set of $m$ clients $\mathcal{I}'(t) \subset \mathcal{I}(t)$ participate training;
5:     Server broadcasts parameters of the network $\boldsymbol{w}^t$ and the $k$-means parameters $(\boldsymbol{c}_1^t, \boldsymbol{c}_2^t, \alpha^t)$ to $\mathcal{I}'(t)$;
6:     **for** clients $i \in \mathcal{I}'(t)$ **in parallel do**
7:         Compute average distance $d_{ik}(\boldsymbol{w}^t) = \frac{1}{|\mathcal{D}_i|} \sum_{\boldsymbol{x} \in \mathcal{D}_i} \alpha_k \|f(\boldsymbol{w}^t, \boldsymbol{x}) - \boldsymbol{c}_k\|$;
8:         Choose the branch with minimal distance $k_i^* = \arg\min_k d_{ik}(\boldsymbol{w})$;
9:         Given $k_i^*$, run local updates for the network by optimizing Eq. 4 and get $\tilde{\boldsymbol{w}}_i^t$;
10:        Compute $q_i(t) = \mathbb{1}[d_{i1}(\tilde{\boldsymbol{w}}_i^t) < d_{i2}(\tilde{\boldsymbol{w}}_i^t)]$ to be aggregated for $\bar{q}(t)$;
11:        Compute $\bar{f}(\tilde{\boldsymbol{w}}_i^t) = \frac{1}{|\mathcal{D}_i|} \sum_{\boldsymbol{x} \in \mathcal{D}_i} f(\tilde{\boldsymbol{w}}_i^t, \boldsymbol{x})$ to be aggregated for $\boldsymbol{c}_1^{t+1}$ or $\boldsymbol{c}_2^{t+1}$;
12:     Server receives aggregated network and $k$-means parameters $\{\bar{\boldsymbol{w}}^t, \boldsymbol{c}_1^{t+1}, \boldsymbol{c}_2^{t+1}, \bar{q}(t)\}$ from the clients;
13:     Update the distance scalar $\alpha_2^{t+1}$ with the temporal prior $\tilde{q}(t)$: $\alpha_2^{t+1} = \exp\left\{\eta^t \left[\tilde{q}(t) - \bar{q}(t)\right]\right\} \alpha_2^t$;
14:     Update the $k$-means centers into $\boldsymbol{c}_1^{t+1}, \boldsymbol{c}_2^{t+1}$ using Eq. 12;
15:     Update network parameters to $\boldsymbol{w}^{t+1}$ using $\bar{\boldsymbol{w}}^t - \boldsymbol{w}^t$ with the prescribed server optimizer.

---

### 3.3 COMPARISONS AND DISCUSSIONS

**Privacy.** In our simulation experiments (Section 4), FEDTEM often performs better than FEDTKM. However, the server has to observe the GMM parameters for participating client in FEDTEM, which may require new techniques to satisfy the strong data minimization and data anonymization principles (Wang et al., 2021; Bonawitz et al., 2021). On the other hand, FEDTKM only uses aggregated results, which is compatible with the strong data minimization principle, and easier to be further protected by other privacy techniques such as secure aggregation (Bonawitz et al., 2019) and differential privacy (McMahan et al., 2018; Kairouz et al., 2021).

**Differences from previous methods.** Compared with previous methods that jointly train a clustering model with multiple networks, our method has four key differences that better suits cross-device FL under temporal distribution shift. First, our method introduces the temporal prior to regularize the clustering model, which is missing in previous works. Second, our $k$-branch network with shared feature extractor is more communication and data efficient than using $k$ networks. Third, our clustering model selects the branches based on the feature, which eliminates the need for labeled data for clients not participate in training. Clustering models of (Ghosh et al., 2020; Marfoq et al., 2021) are based on the loss, which is impractical for clients without labeled data. Fourth, our method does not maintain states on clients. (Marfoq et al., 2021) assumes stateful clients, which is impractical for cross-device setting where each client only participate limited times (Wang et al., 2021). We give more detailed comparison with previous methods in Appendix A.7.

## 4 EXPERIMENTS

**Dataset.** We consider two image classification datasets, EMNIST and CIFAR, and one next word prediction task on Stack Overflow (SO). The split of the day mode ($\mathcal{I}_1$) and night mode ($\mathcal{I}_2$), and other statistics, are shown in Table 1 in the Appendix. On SO, we use a vocabulary size of 10K and report the test accuracy without special tokens. We truncate each sentence to have no more than 20 tokens. For the distribution shift, we use $T = 256$ for the majority of our results, and we compare results under both linear and cosine distribution shifts with $p \in \{0.1, 0.25, 0.5, 1, 2, 4, 10\}$.

**Architecture of the multi-branch networks.** On EMNIST, we train LeNet with 2 Conv layers and 2 FC layers. On CIFAR, we train a ResNet-18 with the Batch Norm replaced by Group Norm (Wu

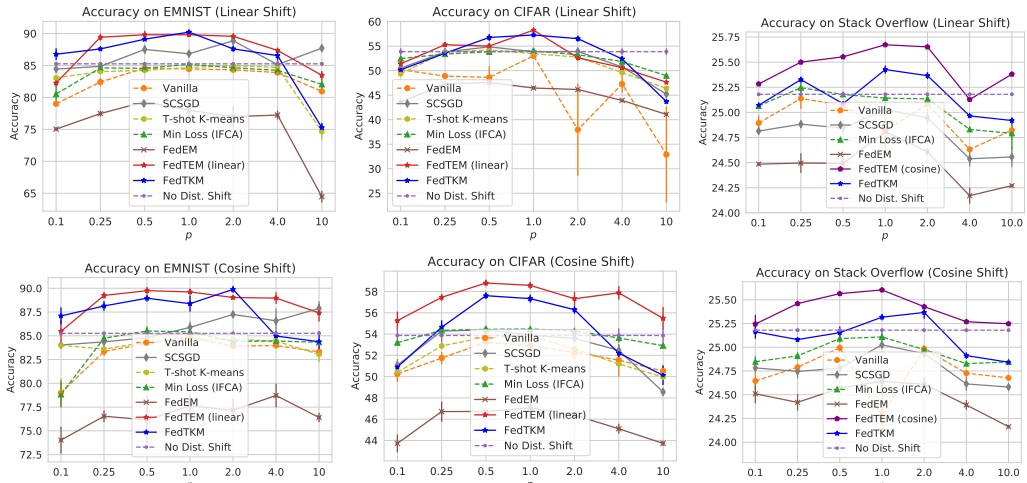

Figure 2: Comparing the results of FEDTEM and FEDTKM with baseline methods. On EMNIST and CIFAR, FEDTKM can achieve similar performances as FEDTEM and outperforms most of the other methods. On Stack Overflow, FEDTKM is not as good as FEDTEM, but it still outperforms other methods, and can be even better than the model trained without distribution shift when the training population is less biased ($p$ is closer to 1).

& He, 2018) for stability in federated learning. We use the last FC layer of these convolutional networks as the multi-branch part and share all the remaining layers. We use the output from the shared feature extractor for clustering, which are 128 and 512 dimensional for EMNIST and CIFAR respectively. On SO, we train a single-layer LSTM (Hochreiter & Schmidhuber, 1997) with a hidden size of 670 and embedding size of 96. To alleviate the communication overhead and prevent overfitting, we define the branches to be the last projection layer before the final prediction layer, which is 15x smaller than the final prediction layer. The communication overheads of the extra branch are 0.7%, 0.5% and 1.6% respectively for models on EMNIST, CIFAR and SO.

**Hyperparameters.** We use FedAdam (Reddi et al., 2021) as the optimizer. Table 2 shows the training hyperparameters, with implementation details and practical notes given in Section A.4. For Min Loss, FEDTEM and FEDTKM, we also do a grid search on the label smoothing parameters $\epsilon$ and $\lambda$. For each hyperparameter setting, we run 3 experiments with different random seeds and report their mean and standard error.

**Ablation study and temporal prior.** In Appendix A.5 and Figure 3, we provide ablation studies on the temporal prior to quantify its efficacy, and see which temporal prior (Linear, Cosine, or Soft) works better in practice. From the results we can see: 1) all three types of priors improve the baseline in most cases; 2) linear prior works better on image datasets, while cosine prior works better on SO; 3) with weaker assumptions, Soft prior is able to fit the distribution shift and improves the results. With these observations, for FEDTEM, we choose the linear prior $\tilde{q}_{L,1}(t)$ on EMNIST and CIFAR and the cosine prior $\tilde{q}_{C,1}(t)$ on SO. For FEDTKM, we only consider the linear prior $\tilde{q}_{L,1}(t)$.

**Baselines.** For fair comparisons, unless otherwise stated, we adapt all methods into our settings to maintain two important properties: 1) training the same multi-branch network to keep model capacities the same; 2) labels are not available for test clients. For the baseline without any branch selection technique, we train the same multi-branch network by taking the averaged output from both branches for prediction. Such networks trained under shifting data distribution are denoted as "**Vanilla**", while those trained without distribution shift are denoted as "**No Dist. Shift**" (NDS).In Appendix A.8, we also provide results of training single-branch models in these two settings, which obtained almost identical results. The baseline methods are: 1) "**SCSGD**", where similar as Semi-Cyclic SGD (Eichner et al., 2019), we train one model during first half of each period, and the other model on the other half. During test time, since there is no indicator of which mode the clients come from in practice, we select the models according to the certainty of their predictions, measured by evaluating the KLD between the prediction and a uniform distribution over all labels. 2) **T-shot K-means**, which is an enhanced variant of one-shot K-means (Dennis et al., 2021), collecting the cluster centers on raw data from all participating clients during a whole period of $T$ rounds before training. For both training and test, it selects the branches for each sample according to the nearest cluster center. 3) "**Min Loss (IFCA)**", where same as IFCA (Ghosh et al., 2020), we choose the

branch with minimum loss on the local training set of each client during training, but different from IFCA, we choose branches with highest certainty on the unlabeled test set clients, using the same criterion as 1). We applied the label smoothing regularization (Eq. 4) to IFCA, which improves the results. 4) **FedEM** (Marfoq et al., 2021), where we use the same algorithm on participating clients during training. FedEM is implemented with privilege to access sample labels for test clients, to enable its EM steps which uses the loss.

**Main Results.** The results of all methods are shown in Figure 2. By comparing results of Vanilla and NDS, we find the temporal distribution shift indeed degrades vanilla training, causing the worst-case decrease in accuracy by more than 7%, 20% and 1% respectively on EMNIST, CIFAR and SO. While with FEDTEM and FEDTKM, the accuracy can be even higher than models trained in the NDS setting in most cases, demonstrating the efficacy of the temporal prior for learning more distinctive feature representations for the daytime and nighttime modes with specialized prediction branches. The improvement on the strong NDS baseline can be as high as 4%, 4% and 0.45% with FEDTEM on the three datasets. FEDTEM is not better than NDS when $p$ deviates too much from 1 as the data distribution is extremely skewed in such settings, while our methods are still better than most other methods in most cases. For baseline methods, SCSGD performs surprisingly well on EMNIST when $p$ is large, but the performance quickly drops on the more complicated CIFAR and SO. T-shot K-means achieves improvements for extreme $p$'s on EMNIST, but it is not significantly better than "Vanilla" on CIFAR, due to the difficulty in reliable clustering in more complicated image spaces. We did not implement T-shot K-means for SO due to the difficulty of clustering the raw data of the language task. Min Loss (IFCA) alleviates the drop and sometimes competes with NDS, but cannot achieve improvements over NDS. By contrast, FedEM often achieves inferior results even with the privilege access to the test labels for EM estimation. This is probably due to the staleness of the priors in the challenging periodic distribution shift setting, as we simulate cross-device FL on a large population where clients only participate training for a few rounds on EMNIST and CIFAR, and at most one round on SO.

**Effect of label smoothing regularization.** Shown in Figure 5. Since EMNIST is relatively easy and the network on it is small, the label smoothing does not show significant improvements. However, it is critical to the success on CIFAR and SO. Without the regularization, one branch will not be updated simultaneously with the feature extractor, resulting in instability and low test accuracy.

**MLE for Evaluation.** For FEDTEM, during evaluation, we find it beneficial for image classification to use the MLE in Eq. 5 on the minibatches (typically of size no larger than 64) to select the best branch, compared with the sample-level inference. We show its benefit in Figure 4 in the appendix. In practice, similar approaches can be realized through caching. We find Min Loss does not show much difference when a similar approach is applied, where branches are chosen by the averaged certainty on minibatches. The baseline model even gets worse performance with this approach, as shown in Figure 4, indicating our model is much better at distinguishing two modes in expectation. However, this approach does not show improvements for FEDTEM on SO.

**Effect of $T$.** As shown in Figure 8, for FEDTEM with linear prior on EMNIST, a smaller $T$ tends to decrease the performance when $p$ deviates too much from 1. In such scenarios, the distribution changes frequently and abruptly. However, the performance of FEDTEM is maintained when the distribution changes in a frequent but more balanced way.

## 5 CONCLUSIONS

In this paper, we showed the influence of a smoothly shifting distribution between daytime and nighttime modes better matches the phenomenon in practical FL systems, and developed algorithms to train multi-branch networks to tackle the distribution shift. Our methods incorporates priors of the temporal distribution shifts to learn a mixture or clustering model to guide the network and select corresponding branches for clients from different modes during both training and inference. The clustering models are defined in the feature space and does not require labelled data for inference, hence is ready to be deployed on clients without labeled data. The branches are lightweight with little communication overhead, and the model demonstrates significant improvements in test accuracy on the benchmark datasets. Our methods also satisfies many other real-world constraints like single round communication and stateless clients (Bonawitz et al., 2019; Paulik et al., 2021; Wang et al., 2021), which makes it possible to be deployed in practical FL systems.

## 6 REPRODUCIBILITY STATEMENT

Our code is available on GitHub.[2] For reproducibility, we have reported the mean and standard error for 3 experiments with different random seeds for all settings. The dataset we used for simulations are all available in Tensorflow Federated, and our implementation is largely based on the code from Adaptive Federated Optimization (Reddi et al., 2021).[3] We have listed our hyperparameters in Table 2. We have given the pseudo code in Algorithm 1 and the detailed formulas and derivations of our algorithm in Section 3.1.

## 7 ACKNOWLEDGEMENTS

The authors would like to thank Galen Andrew and Brendan McMahan for helpful discussions. Goldstein was supported by the National Science Foundation #1912866, the DARPA GARD program, and the Sloan Foundation.

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

# A APPENDIX

## A.1 FURTHER DISCUSSIONS OF BRANCH SELECTION WITH GMM

**Derivation of the MLE.** Recall $\boldsymbol{\theta}$ is the parameters of the GMM model. We consider maximizing the expectation of the complete-data log likelihood $\log P(\mathcal{D}_i, \boldsymbol{Z}|\boldsymbol{\theta})$ under the posterior distribution $P(\boldsymbol{Z}|\mathcal{D}_i, \boldsymbol{\theta})$, where $\boldsymbol{Z}$ is the collection of latent variables associated with each of the sample $\boldsymbol{x} \in \mathcal{D}_i$. Formally, it is solving the following constrained optimization problem

$$\underset{\boldsymbol{\theta}}{\text{maximize}} \; \mathbb{E}_{\boldsymbol{Z} \sim P(\boldsymbol{Z}|\mathcal{D}_i, \boldsymbol{\theta})} \left[ \log P(\mathcal{D}_i, \boldsymbol{Z}|\boldsymbol{\theta}) \right],$$
$$\text{subject to} \; \sum_{j=1}^{K} \pi_j = 1, \pi_j \geq 0, \text{for all } 1 \leq j \leq K. \tag{14}$$

We introduce an indicator function $\mathbb{1}_{[z=k]}$ which is 1 if $z = k$ or 0 otherwise. In this way, the log likelihood can be represented as

$$\log P(\mathcal{D}_i, \boldsymbol{Z}|\boldsymbol{\theta}) = \log \prod_{n=1}^{|\mathcal{D}_i|} \prod_{k=1}^{K} [P(\boldsymbol{x}_n, z_n|\boldsymbol{\theta})]^{\mathbb{1}_{[z=k]}}$$
$$= \sum_{n=1}^{|\mathcal{D}_i|} \sum_{k=1}^{K} \mathbb{1}_{[z_n=k]} \left[ \log \pi_k + \log P(\boldsymbol{x}_n|z_n, \boldsymbol{\theta}) \right]. \tag{15}$$

Meanwhile,

$$\mathbb{E}_{\boldsymbol{Z} \sim P(\boldsymbol{Z}|\mathcal{D}_i, \boldsymbol{\theta})} \left[ \mathbb{1}_{[z_n=k]} \right] = P(z_n = k|\boldsymbol{x}_n, \boldsymbol{\theta}) = \frac{\pi_k P(\boldsymbol{x}_n|z_n = k, \boldsymbol{\theta})}{\sum_{j=1}^{K} \pi_j P(\boldsymbol{x}_n|z_n = j, \boldsymbol{\theta})}. \tag{16}$$

Plug Eq. 15 and Eq. 16 back into Eq. 14, and ignore terms that do not depend on $\boldsymbol{\pi}$, we find we are solving the following problem for $\boldsymbol{\pi}$

$$\underset{\boldsymbol{\pi}}{\text{maximize}} \; \sum_{n=1}^{|\mathcal{D}_i|} \sum_{k=1}^{K} P(z_n = k|\boldsymbol{x}_n, \boldsymbol{\theta}) \log \pi_k,$$
$$\text{subject to} \; \sum_{j=1}^{K} \pi_j = 1, \pi_j \geq 0, \text{for all } 1 \leq j \leq K. \tag{17}$$

The Lagrangian multiplier of this problem is

$$h(\boldsymbol{\pi}, \lambda) = \sum_{n=1}^{|\mathcal{D}_i|} \sum_{k=1}^{K} P(z_n = k|\boldsymbol{x}_n, \boldsymbol{\theta}) \log \pi_k + \lambda(1 - \sum_{j=1}^{K} \pi_j). \tag{18}$$

Let the derivative w.r.t. $\pi_k$ be 0,

$$\frac{\partial h}{\partial \pi_k} = \sum_{n=1}^{|\mathcal{D}_i|} P(z_n = k|\boldsymbol{x}_n, \boldsymbol{\theta}) \frac{1}{\pi_k} - \lambda = 0. \tag{19}$$

Multiply both sides of Eq. 19 with $\pi_k$, sum over $1 \leq k \leq K$, and apply the constraint that $\sum_{k=1}^{K} \pi_k = 1$, we have

$$\lambda = \sum_{n=1}^{|\mathcal{D}_i|} \sum_{k=1}^{K} P(z_n = k|\boldsymbol{x}_n, \boldsymbol{\theta}) = |\mathcal{D}_i|. \tag{20}$$

Plug this back into Eq. 19, we get the equation as desired.

$$\pi_k = \frac{1}{|\mathcal{D}_i|} \sum_{n=1}^{|\mathcal{D}_i|} P(z_n = k|\boldsymbol{x}_n, \boldsymbol{\theta}) \tag{21}$$

**An alternative for branch selection.** Instead of computing the MLE, with the assumption that samples in $\mathcal{D}_i$ are IID, we can compute $P(\zeta_i|\mathcal{D}_i) = P(z|\mathcal{D}_i)$ using Bayes' theorem and use $P(\zeta_i|\mathcal{D}_i)$ for branch selection. Specifically,

$$P(\zeta_i = k|\mathcal{D}_i) = P(z = k|\mathcal{D}_i) = \frac{\pi_k \prod_{\boldsymbol{x} \in \mathcal{D}_i} P(\boldsymbol{x}|z = k)}{\sum_{j=1}^{K} \pi_j \prod_{\boldsymbol{x} \in \mathcal{D}_i} P(\boldsymbol{x}|z = j)}. \tag{22}$$

As shown in Figure 6, we find MLE achieves better empirical results than this alternative.

## A.2 DERIVATION OF THE LOCALLY OPTIMAL GMMS

We give the derivation of the optimal mean and variance of the GMM on each client, under the one-hot assumption that $P(\zeta = k|\mathcal{D}_i, \boldsymbol{\theta}^t, \tilde{q}(t)) = 1$ if $k = k^*$, and $P(\zeta = k|\mathcal{D}_i, \boldsymbol{\theta}^t, \tilde{q}(t)) = 0$ if $k \neq k^*$, for some given $k^*$. Under this assumption, we need to solve

$$\begin{aligned}
\tilde{\boldsymbol{\theta}}_{ik^*}^{t+1} &= \underset{\boldsymbol{\theta}}{\operatorname{argmax}} \ \log P(\mathcal{D}_i, \zeta = k^*|\boldsymbol{\theta}, \tilde{q}(t)) \\
&= \underset{\boldsymbol{\theta}}{\operatorname{argmax}} \ \log P(\zeta = k^*|\boldsymbol{\theta}, \tilde{q}(t)) + \sum_{\boldsymbol{x} \in \mathcal{D}_i} \log P(\boldsymbol{x}|\zeta = k^*, \boldsymbol{\theta}, \tilde{q}(t)) \\
&= \underset{\boldsymbol{\theta}}{\operatorname{argmax}} \ \log \pi_{k^*} - \sum_{\boldsymbol{x} \in \mathcal{D}_i} \left[ \sum_{j=1}^{d} \frac{([f(\boldsymbol{w}_i^{t+1}, \boldsymbol{x})]_j - [\boldsymbol{\mu}_{k^*}]_j)^2}{2 [\boldsymbol{\sigma}_{k^*}]_j^2} + \log [\boldsymbol{\sigma}_{k^*}]_j \sqrt{2\pi} \right]
\end{aligned} \tag{23}$$

where $d$ is the dimension of the features, $[\boldsymbol{\mu}_{k^*}]_j$, $[\boldsymbol{\sigma}_{k^*}]_j$ and $[f(\boldsymbol{w}_i^{t+1}, \boldsymbol{x})]_j$ denote the $j$-th dimension of $\boldsymbol{\mu}_k, \boldsymbol{\sigma}_k, f(\boldsymbol{w}_i^{t+1}, \boldsymbol{x})$, respectively. From Eq. 23, we can see the optimal solution for client $i$ is to set the mean and variance for the $k^*$-th mode as

$$\tilde{\boldsymbol{\mu}}_{ik^*}^{t+1} = \frac{1}{|D_i|} \sum_{\boldsymbol{x} \in \mathcal{D}_i} f(\boldsymbol{w}_i^{t+1}, \boldsymbol{x}), \ [\tilde{\boldsymbol{\sigma}}_{ik^*}^{t+1}]^2 = \frac{1}{|\mathcal{D}_i|} \sum_{\boldsymbol{x} \sim \mathcal{D}_i} (f(\boldsymbol{w}_i^{t+1}, \boldsymbol{x}) - \tilde{\boldsymbol{\mu}}_{ik^*}^{t+1})^2. \tag{24}$$

## A.3 THE SOFT PRIOR

The soft prior $\tilde{q}_S(t)$ gives strong signals only at time steps where we believe $q(t) = 0$ or $q(t) = 1$. In between, it only requires the ratio to be non-increasing or non-decreasing, where the ratio is estimated using the posterior from Eq. 6 as

$$\tilde{q}_S'(t) = \frac{|\{\boldsymbol{x}|\boldsymbol{x} \in \mathcal{D}_i, \gamma_{i1}(\boldsymbol{x}) > \gamma_{i2}(\boldsymbol{x})\}|}{|\mathcal{D}_i|}, \tag{25}$$

i.e., the ratio of samples whose posterior has higher probability on mode 1. In this way, the model can still learn to distinguish the modes but we do not enforce a strong prior at every time step. Specifically, it is defined as

$$\tilde{q}_S(t) = \begin{cases} 1, & \text{if } t \bmod T = 1 \\ 0, & \text{if } t \bmod T = \frac{T}{2} \\ \min(\tilde{q}_S(t-1), \tilde{q}_S'(t)), & \text{if } 1 < t \bmod T < \frac{T}{2} \\ \max(\tilde{q}_S(t-1), \tilde{q}_S'(t)). & \text{if } t \bmod T > \frac{T}{2} \end{cases} \tag{26}$$

## A.4 IMPLEMENTATION DETAILS AND PRACTICAL NOTES

**Setting the GMM prior.** During test, to compare the model performance with established baselines and ensure fairness, we consider a static test set where the number of clients and samples from both modes are roughly the same. Therefore, we use a uniform distribution for the priors on the test set. This does not make the GMM worse if it estimates $P(\boldsymbol{x}|z)$ accurately on the test set. During training, we estimate the prior $\boldsymbol{\pi}^t$ in every step to match the shifting distribution. However, we find it beneficial to use a running average of $\pi_k^{t+1} = \beta\pi_k^t + (1-\beta)\pi_k M_k^t/(\sum_{j=1}^{K} M_j^t)$ during training, which achieves better results than setting $\beta = 0$ or using uniform prior in our preliminary experiments. We use $\beta = 0.99$ for the moving average in all experiments.

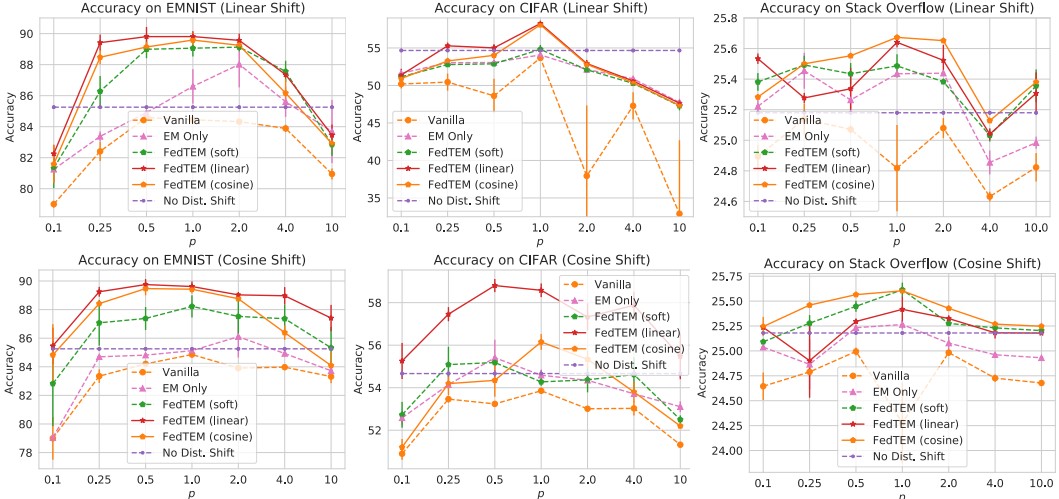

Figure 3: The effect of the temporal prior. Note we always use the same periodical linear or cosine priors, $\tilde{q}_{L,1}(t)$ or $\tilde{q}_{C,1}(t)$, under all types and $p$'s for the distribution shift.

**Setting $\eta_t$ for FEDTKM.** Although we can identify the modes through the local minima and maxima of the training curve, the distribution shift can happen in arbitrary ways in between. As a result, the certainty of the temporal prior should also be lower in between the modes. To incorporate this heuristic, instead of using a constant $\eta^t$, we let

$$\eta^t = 2|0.5 - \tilde{q}(t)|\eta_{\max}, \text{ where } \eta_{\max} > 0 \text{ is a constant.} \tag{27}$$

We find this to give better results when $p \neq 1$, i.e., when the underlying distribution shift is biased and different from the linear temporal prior.

## A.5    ABLATION STUDIES: THE EFFECT OF TEMPORAL PRIOR

To quantify the effect of temporal prior on training, we compare four versions of FEDTEM: 1) **EM Only**: only apply the EM part of our algorithm without the temporal priors, where the only difference from FEDTEM is that the server, in the M step, greedily uses client $i$ to update parameters of mode $k_i^* = \arg\max_k \tilde{\pi}_{ik}^*$ (see Eq. 9 and Eq. 10); 2) **FedTEM (linear)**: our algorithm where the temporal prior is the periodic linear function $P(i \in \mathcal{I}_1(t)) = q_{L,1}(t)$; 3) **FedTEM (cosine)**: our algorithm where the temporal prior is the cosine function $P(i \in \mathcal{I}_1(t)) = q_{C,1}(t)$; 4) **FedTEM (soft)**: our algorithm with the soft temporal prior introduced in Section A.3. The results are shown in Figure 3. We find all three priors improves the "EM Only" baseline in most cases. "FedTEM (linear)" is better on image datasets, while "FedTEM (cosine)" is better on the language dataset. The soft prior has very weak assumptions about the prior, only requiring the estimated ratios to be non-increasing or non-decreasing within certain intervals, but still improves the results.

Table 1: Stats of the datasets. We use $|\mathcal{D}_.|$ to denote number of samples in each subset. On SO, # Classes is the vocabulary size.

| Dataset | # Classes | $\mathcal{I}_1$ | $|\mathcal{I}_1|$ | $|\mathcal{D}_{\mathcal{I}_1}|$ | $\mathcal{I}_2$ | $|\mathcal{I}_2|$ | $|\mathcal{D}_{\mathcal{I}_2}|$ | $|\mathcal{D}_{\text{test}}|$ |
|---|---|---|---|---|---|---|---|---|
| EMNIST | 62 | Digits | 3383 | 341,873 | Characters | 3400 | 329,712 | 77,483 |
| CIFAR | 110 | CIFAR10 | 500 | 50,000 | CIFAR100 | 500 | 50,000 | 20,000 |
| Stack Overflow | 10K | Questions | 171K | 67M | Answers | 171K | 67M | 16M |

## A.6    VERIFYING THE EFFECT OF MLE FOR EVALUATION

To quantize the effect the batch-level MLE, we compare the results of our method with or without MLE on EMNIST and CIFAR. As shown in Figure 6, this only decreases the accuracy of the baseline model, indicating our model is much better at distinguishing the two modes in expectation.

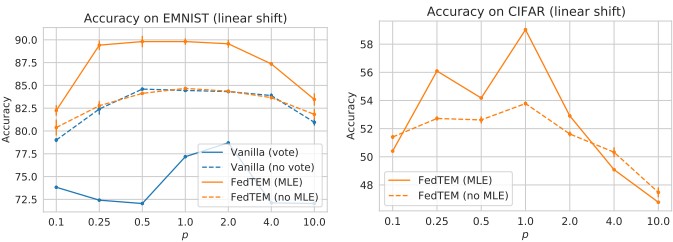

Figure 4: Comparing the effect of MLE on EMNIST and CIFAR.

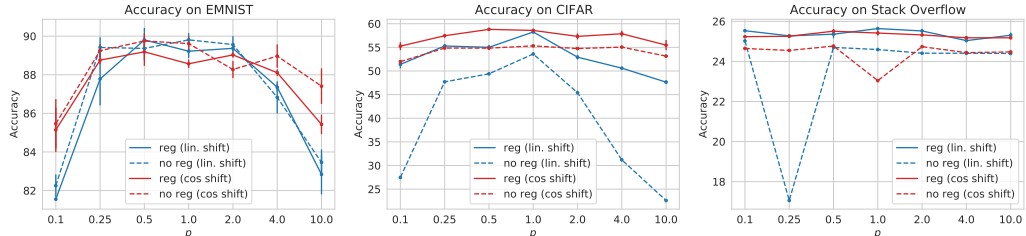

Figure 5: Compare the effect of the label smoothing regularization on FEDTEM under linear and cosine distribution shifts. We use the linear prior in all cases.

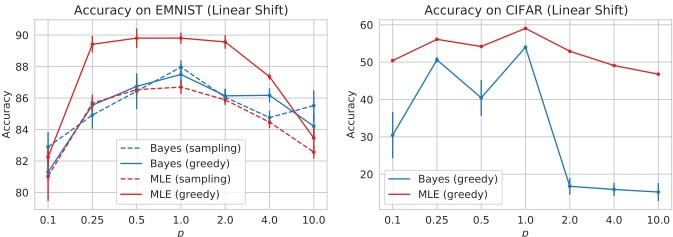

Figure 6: Comparing the Bayesian and MLE based branch selection on EMNIST and CIFAR under linear distribution shifts. We also compare the greedy branch selection vs. sampling based branch selection on EMNIST.

Table 2: Training hyperparameters on the datasets. The network parameters are trained for one epoch on each client. In addition, on Stack Overflow, we also limit each client to use no more than 512 samples during training.

| Dataset | Server Opt | Server LR | $\epsilon$ | Client Opt | Client LR | $|\tilde{I}(t)|$ | Batch Size | Total Rounds |
|---|---|---|---|---|---|---|---|---|
| EMNIST | Adam | $10^{-2.5}$ | $10^{-4}$ | SGD | $10^{-1.5}$ | 10 | 20 | 2049 |
| CIFAR | Adam | 1 | $10^{-1}$ | SGD | $10^{-1.5}$ | 10 | 20 | 8196 |
| Stack Overflow | Adam | 0.01 | $10^{-5}$ | SGD | $10^{-0.5}$ | 50 | 16 | 2048 |

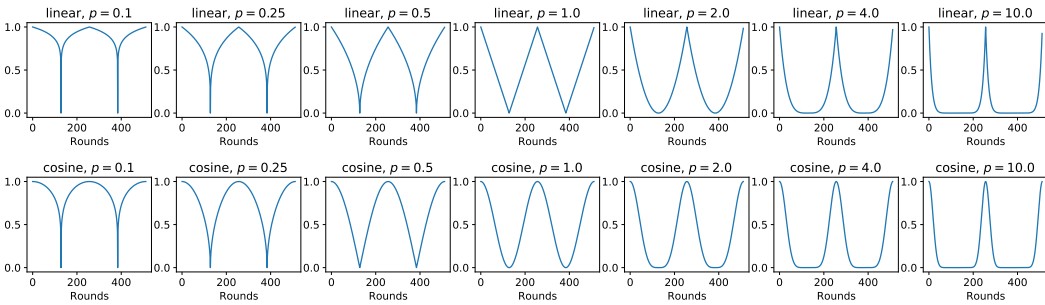

Figure 7: Probability of sampling of clients from $\mathcal{I}_1(t)$ in each round, with $T = 256$, under the linear and cosine transit functions and different values of $p$.

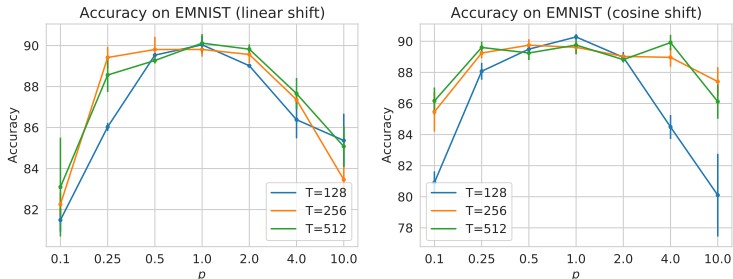

Figure 8: Evaluating the effect of the underlying $T$ on FEDTEM with linear prior.

## A.7    ADDITIONAL RELATED WORKS

As mentioned above, Semi-cyclic SGD (Eichner et al., 2019) and (Ding et al., 2020) are the only previous works we are aware of that explicitly consider periodical distribution shift in FL. We now review other related works that either consider other types of distribution shift, or have (weak) similarity as the proposed FEDTEM method.

**Heterogeneity and client distribution shift.**    Client heterogeneity is an active topic in FL, and various methods have been proposed to address the distribution difference among clients. Notably, FedProx (Li et al., 2020b) applies proximal regularizer when perform client updates; SCAFFOLD (Karimireddy et al., 2020) uses control variates as local states on clients for variance reduction; FedPA (Al-Shedivat et al., 2021) provides a Bayesian view and adopts posterior sampling; FedGen (Zhu et al., 2021) learns a generator that can be shared among clients; FedRobust (Reisizadeh et al., 2020) applies gradient descent ascent to tackle distribution shifts in the form of affine transforms in the image domain. Transfer learning, multi-task learning, and meta learning are introduced into FL to explicitly handle distribution shifts among clients assuming heterogeneous clients are from different domains or tasks (Smith et al., 2017; Khodak et al., 2019). Continual learning is introduced to handle distribution shift due to streaming tasks on each client (Yoon et al., 2021). More recently, distribution shift in clusters of clients instead of each individual client are studied in Mansour et al. (2020); Ghosh et al. (2020), while still assuming the clients can be uniformly accessed during the training process. We kindly ask interested readers to find more papers on heterogeneity in a few recent surveys (Kairouz et al., 2019; Li et al., 2020a; Wang et al., 2021). All these methods consider the distributions shift among different clients, while the proposed FEDTEM considers periodic distribution shift of client population.

**Clustering and mixture models.**    We do not assume strong control over the clients and the available client population is always changing due to the periodical distribution shift. We also do not require stateful clients, since our prior is applied globally to all clients. Our mixture model is based on the feature space, therefore we do not require labeled data for unseen clients to infer its mode. To our knowledge, existing works in clustered FL, including those on personalization, mostly fail to satisfy at least one of the three properties and therefore not applicable in our setting. Dennis et al. (2021) propose a one-shot clustering approach based on the raw client data before training, which implicitly assumes all representative clients are available simultaneously. Clustering on raw data can also be unreliable when the data demonstrates complicated distributions in the input space. Clustered Federated Learning (Sattler et al., 2020) applies an one-shot clustering based on a trained global model, and then train a personalized model for each cluster. To achieve this, it implicitly assumes strong control over population and clients sampling. Ghosh et al. (2020) and Mansour et al. (2020) propose a similar algorithm that alternatively perform clustering and updating the personalized models for corresponding clusters. Both of them require labled data for the clients to compute the loss for model selection. During the preparation of this draft, we notice a concurrent work, FedEM (Marfoq et al., 2021), which proposes a Federated EM algorithm to learn a mixture model for each client and weigh the predictions from multiple models. The modified EM algorithm requires computing the loss and therefore labeled data for every client. It maintains a different prior distribution for every client, which requires stateful clients and strong control over client sampling. The hierarchical or bi-partitioning clustering process of (Briggs et al., 2020; Fu et al., 2021) requires all

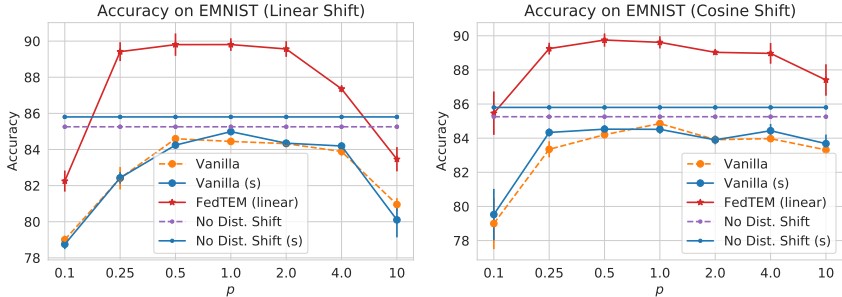

Figure 9: The effect of using multi-branch networks for the baselines. "Vanilla (s)" is the single-branch network trained with FedAdam under various distribtuion shifts. "No Dist. Shift (s)" is the single-branch network trained with FedAdam with no distributoin shift.

clients to be available for clustering simultaneously, violating our assumption that the distribution of the population is constantly changing and is not practical in on-device FL in the real world. In addition, for unseen clients, either additional communications are needed for determining their clusters, or the client has to download models for all clusters, which adds significantly to the communication cost. (Xie et al., 2021) maintains one set of weights on each client while maintaining multiple weights as cluster centers on the server. It takes two communication rounds for each weight update, which can be impractical. It remains unclear if this mechanism can be generalized to unseen clients. (Marfoq et al., 2021) trains a different set of weights for each node/client, which requires all the nodes/clients to be available all the time. It is more suitable for the cross-silo setting but not practical for on-device setting. (Duan et al., 2021) clusters the clients based on the similarities between their gradients at the initial model weights. This is impractical in our setting since the training population changes throughout the day and we cannot assume clients from both modes are available at the first round for clustering. In addition, there is no guarantee that the gradient at the chosen round can separate the clusters well. And to compute the gradients, it still requires labeled data. (Andreux et al., 2020) advocates using different sets of BNs to track different running statistics for different silos, which is not practical for on-device FL since BN has been widely observed to cause instability in on-device FL, and is often replaced by Group Normalization which does not track running statistics (Hsieh et al., 2020; Hsu et al., 2020). Another limitation is it assumes every client is seen during training.

**Multi-branch networks in FL.** Multi-branch networks have been explored in FL when a personalized model is preferred for each client: the branchs are either locally stored on clients (Arivazhagan et al., 2019; Liang et al., 2020), or reconstructed based on client data (Singhal et al., 2021). All the aforementioned three works require labeled data and additional training efforts to learn the weights of the branches. By comparison, our method does not need re-training the branches and does not need labeled data for unseen clients to select the branches. The branch selection is based on the mixture model in the feature space, and the weights of the branches are fixed.

## A.8 COMPARING SINGLE-BRANCH AND MULTI-BRANCH BASELINES WITHOUT CLUSTERING

For fair comparisons, we have used two branches for the baselines ("Vanilla" and "No Dist. Shift") so that their capacities are the same as our method. Since these two baselines are just training the networks with FedAdam, it is more natural to train single-branch networks directly. In Figure 9, we show that the single-branch and multi-branch networks obtain similar results under various settings, with or without distribution shift. Our method still outperforms these two single-branch baselines.

## A.9 THE EFFECT OF LOCAL TRAINING SET SIZES

When the number of training samples is small, the EM estimates may have high variance, causing misspecifications and resulting in worse performance. Meanwhile, models trained on smaller training sets tend to have worse generalization, so the test accuracy will drop for any method in general. To see which factor has more significant effect on the test accuracy, we compare with the

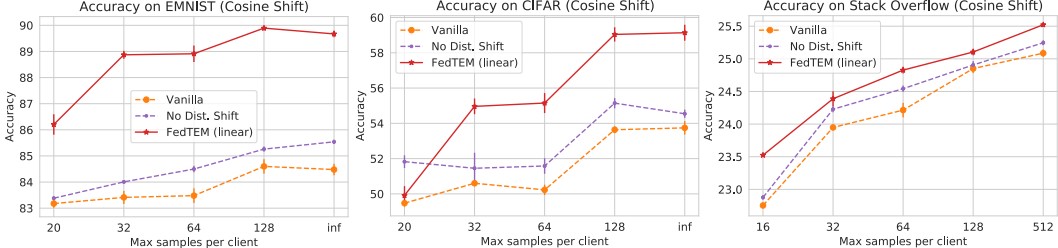

Figure 10: The effect of local training set sizes of each client. x-axis represents the maximum number of training samples per client. For "inf", we use all available training samples from the original dataset. To add to the challenge, we ensure the distribution shift is different from the temporal prior. On EMNIST and CIFAR, we consider cosine data distribution shift with $p = 1$ and use linear prior for FedTEM. On Stack Overflow, we consider cosine distribution shift and linear prior. We also compare with the results for the baseline model trained without distribution shift (No Dist. Shift).

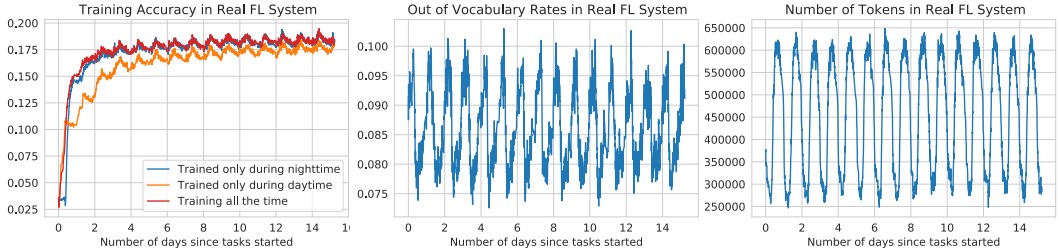

Figure 11: The training accuracy, out of vocabulary rates and total number of tokens at each round for training a next word prediction model in a real FL system. In general, the out of vocabulary rates become lower on nighttime clients, and the sentence lengths become longer on nighttime clients. The plots also show training is faster during nighttime, since more rounds are finished during nighttime.

baselines trained with or without distribution shift in Figure 10, where we only change the maximum training samples per client while keeping other hyperparameters unchanged. From the plot, we can see FedTEM maintains the advantage over the baselines and oracle (No Dist. Shift) under various training set sizes, though its accuracy also decays as the baselines due to the decreased numbers of training samples.

## A.10   STATS OF TRAINING A LANGUAGE MODEL IN A PRACTICAL FL SYSTEM

In Figure 11, we show some stats that could characterize the data distribution shifts in a real FL system. Generally, the data of daytime clients is more difficult to fit and generalize. Data during daytime has higher out of vocabulary rates, and lower sequence length, indicating these sentences might be more arbitrary or informal than data during nighttime. From the plots, we can also see more rounds are completed during nighttime. This is because the product is mainly deployed in one region, so during daytime, fewer devices are idle for training and the round completion rates are lower (Yang et al., 2018). These plots further justify that the data shifts smoothly rather than abruptly in practice.

