# OpenReview forum: "Diurnal or Nocturnal? Federated Learning of Multi-branch Networks from Periodically Shifting Distributions"
_ICLR.cc/2022/Conference — ICLR 2022 Poster_

### Official Review · Reviewer_JqZ9 · 2021-11-01

**Correctness:** 4
**Technical Novelty And Significance:** 3
**Empirical Novelty And Significance:** 4
**Recommendation:** 8
**Confidence:** 2

**Main Review:**

# Strengths

- Well written and clear paper, with a good motivation (fig 1)
- The proposed method is novel to the best of my knowledge, and provides an empirical solution to the problem initially tackled by Eichner et al.
- Although the authors focus on the case of 2 components in the experiments, the method is quite generic and could be applied in different settings
- The experiments are run on different datasets, with multiple relevant ablation baselines, and the results demonstrate the impact of the proposed method as well as the resilience to errors in modelisation (e.g. cosine instead of linear)

# Weaknesses

- In the experiments, the baselines are not that natural insofar as they also have multiple heads. I think it would also be relevant to compare to a network with a single branch.
- Fig 2 is difficult to read due to the large number of curves in each plot: I suggest adding different markers to make it more colorblind-friendly.
- The temporal prior is not learnt but fixed. Do you think it would be possible to fit it as well from data?
- Only 3 seeds are used: in addition to the means and stds, it would be great to provide the individual points, or, alternatively, to use more seeds.

**Summary Of The Paper:**

This paper investigates the problem of periodic client distribution shift in cross-device FL, as previously investiged by Eichner et al (semi-cyclic SGD). The authors propose to split the NN model in a shared part, with different heads for each component. A GMM model is added on top of the output of the shared part to allocate each client to a given head during training, with a temporal prior for the shift between classes. This GMM is fit via a federated EM scheme, in addition to the federation of the NN weights themselves. Experiments are conducted on EMNIST, CIFAR10/100 and StackOverflow datasets, for 2 components. The results demonstrate the superiority of the proposed method with respect to baseline approaches, including the addition of the temporal prior.

**Summary Of The Review:**

Given the technical contributions as well as the well-conducted experiments and its overall quality, I am in favour of accepting this paper.

---

> ### Author Response · Authors · 2021-11-17
> **Response to Reviewer JqZ9**
>
> Many thanks for acknowledging the motivation of our setting, the novelty of our method and the significance of our results. We appreciate your constructive feedback and have added discussions and experiments accordingly.
>
> ### 1. “I think it would also be relevant to compare to a network with a single branch.”
>
> **We have included single-branch baselines with or without distribution shift on EMNIST in Figure 9 of the new version.** We were using multi-branch baselines for fair comparisons, since multi-branch networks should perform better due to a slightly larger number of trainable parameters.  From the new results, we can see that the two versions do not have significant differences, and our method outperforms both. We will add results for other datasets soon.
>
> ### 2. Adding different markers to Figure 2 to make it more colorblind-friendly
> Thank you for the suggestions! We have edited it to make it colorblind-friendly.
>
>
> ### 3. Is it possible to fit the temporal prior from data?
>
> **The “soft” prior we tried in the paper is trying to fit the underlying distribution shift with minimum constraints on the EM model.** It only assumes that the distribution is non-increasing or non-decreasing within certain intervals, and only modifies the assignments when the assumptions are violated. Our experiments show it can also be effective in most cases on EMNIST and Stack Overflow. We think improving the ability to fit the underlying data distribution shift is an important future work.
>
> ### 4. Only 3 seeds are used: in addition to the means and stds, it would be great to provide the individual points, or, alternatively, to use more seeds.
>
> Thank you for your suggestions! Since the current plots are already very crowded, we plan to re-run the experiments with more seeds in our final version.

---

### Official Review · Reviewer_QWqv · 2021-11-03

**Correctness:** 3
**Technical Novelty And Significance:** 2
**Empirical Novelty And Significance:** Not applicable
**Recommendation:** 5
**Confidence:** 5

**Main Review:**

Strengths:

1. The problem setting is interesting and new.
2. It is interesting studying the periodic distribution shift problem in FL. The authors model the distribution shift with the prior daytime and nighttime modes in daily life.
3. The authors propose a novel federated expectation-maximization algorithm (FedTEM) to learn a branched neural network model. Experiments on simulated environments show the proposed method achieves better performance when distribution shifts in the federation.


Weaknesses:

1. The periodic distribution shift problem in FL is very interesting. The distribution shift problem usually expects uncertain changes in every shift step. However, the targeting problem in this paper is a shift between two fixed distributions in a periodic manner. This setting is much less challenging than other distribution shift problems.
2. The experiment cannot well support the mentioned application scenarios on “Diurnal or Nocturnal”. It simulates two distributions in FL benchmark datasets that are not designed to distinguish daytime and nighttime.
3. The baseline selection and experimental comparison is weak.


Questions:

1. From an application perspective, there are many straightforward solutions to solve the proposed problem in training two models for “Diurnal or Nocturnal” respectively. For example, we can train two global models in a totally separate mode. I am not convinced that the proposed solution is necessary to solve the targeting problem. It would be much help to give more detailed examples to enhance the motivation.
2. Could you please provide more details about the difference between a daytime distribution and nighttime distribution? For example, how the distribution changes with respect to p(y), p(x) or p(y|x) ?
3. The proposed FedTEM estimates the mode a client belongs to and later chooses the corresponding branch of a neural network model for it. However, a similar strategy has been reported in the clustered FL framework as mentioned in A.4 - Clustering and mixture model, and other related methods, such as the below papers. Please pick up some paper as baseline methods to compare with.

https://arxiv.org/abs/2004.11791

https://arxiv.org/abs/2108.08647

https://arxiv.org/abs/1705.10467

https://arxiv.org/abs/2103.00697

https://arxiv.org/abs/2108.09749

https://link.springer.com/chapter/10.1007/978-3-030-73194-6_3

https://link.springer.com/chapter/10.1007/978-3-030-60548-3_13




**Summary Of The Paper:**

The paper is to solve the distribution shift between daytime modes and nighttime modes in federated learning with a mixture of distributions. The proposed method could be viewed as a two-group clustered federated learning method. In particular, the client clustering is based on the model parameters of prediction layers.

**Summary Of The Review:**

In this paper, the authors propose a novel setting for federated learning, and then design a solution to handle the periodic distribution shift problem in FL. However, the claim is not well supported by the contents.

---

> ### Author Response · Authors · 2021-11-17
> **Response to Reviewer QWqv (1)**
>
> Thank you for acknowledging the importance of studying periodic distribution shifts in FL, and the novelty of our algorithm. We have added three more methods for comparisons in our experiments (Semi-cyclic SGD [Eichner et al., 2019], One-shot K-means [Dennis et al., 2021], and FedEM [Marfoq et al., 2021]), in addition to the IFCA (Min Loss) [Ghosh et al., 2020] baseline in the previous version. See Figure 2. Note that we have to adapt a variant for Semi-cyclic SGD, T-shot K-means, and FedEM as they cannot directly apply to the cross-device FL setting with client sampling, unseen test clients and periodical distribution shifts.
>
> ### 1. *“...the targeting problem in this paper is a shift between two fixed distributions in a periodic manner. This setting is much less challenging than other distribution shift problems.”*
>
>
> The periodic distribution shift provides extra information that can potentially help us design stronger algorithms that can not be applied to the worst-case distribution shift scenario as you suggested, where the distribution shifts in every step. However, as shown in Figure 1, periodic distribution shift is a real problem in practice, which makes developing specific algorithms for this setting an interesting problem. **We have compared multiple baselines that were designed for general settings without considering periodic distribution shifts, including FedAvg, One-shot K-means, IFCA and FedEM, and the proposed FedTEM is consistently better.** We are happy to discuss and compare more if the reviewer can provide more detail on “other distribution shift problems”
>
> In the context of Federated Learning, as discussed in Appendix A.5, distribution shift is often discussed in the context of client heterogeneity. Addressing client heterogeneity is an active research direction. The proposed FedTEM is compatible with these methods. **Within our setting, inter-client distribution shift still exists like many other papers**, e.g., the local datasets of each client in EMNIST and Stack Overflow datasets are naturally generated by different users with different writing styles or ways of speaking. **Our setting adds additional challenges compared to other papers studying client heterogeneity in FL.** Please refer to our updated related works section for detailed comparisons with existing works.
>
> ### 2. *“It simulates two distributions in FL benchmark datasets that are not designed to distinguish daytime and nighttime.”*
>
> **Our simulation does capture important properties of the distribution shift in practical FL systems, as shown in Figure 1.** Many works in Federated Learning also create local training sets of the clients from similar datasets that are not naturally divided into such subsets. We leave collecting data from real daytime and nighttime users as an important future work.
>
> ### 3. *“.. there are many straightforward solutions to solve the proposed problem in training two models for “Diurnal or Nocturnal” respectively…. we can train two global models in a totally separate mode. ”*
>
> We appreciate it if the reviewer can elaborate more on the straightforward solutions. **To our understanding, the suggested method is very similar to Semi-cyclic SGD [Eichner et al. 2019].** They assume the client switches between two distributions, one for the daytime and nighttime, which can be easily leveraged by training one model during daytime and the other one during nighttime as you suggested. However, **it is not appropriate for our smooth transition setting, since the participating clients are mostly a mixture of both modes and we cannot guarantee each model specialize at one mode by training it only during daytime or nighttime.**
>
> We added a variant of semi-cyclic SGD in our experiments, which does not assume the mode of the clients are known, and selects the models according to the certainty of their predictions during test time. It trains one model during the daytime (first half of each period) and the other model during nighttime (second half of each period). The results are in Figure 2. **This method achieved some surprisingly good results on EMNIST when the data distribution is extremely biased and better matches its model selection strategy during training, but the results on the more challenging CIFAR and Stack Overflow datasets are not as good as our method.**
>
> Regarding using multi-branch instead of multi-model, the advantage of using multi-branch models instead of multiple models is two-fold. First, it reduces the communication overhead, which is critical for cross-device FL [Wang et al., 2021]. For $K$ clusters, we only need to add $(K-1)$ extra linear layers, while the multi-model approach will need to send $(K-1)$ more copies of the whole model. **On EMNIST, CIFAR and Stack Overflow, the communicational overheads are 0.7%, 0.5% and 1.6%**, respectively. Second, it improves data efficiency, in that data from all clusters are used to jointly train the feature extractor.

---

> ### Author Response · Authors · 2021-11-17
> **Response to Reviewer QWqv (2)**
>
> ### 4. More details about the difference between a daytime distribution and nighttime distribution
>
> It is a non-trivial task to characterize the data distribution in the real world, and it is even more difficult to characterize the data distribution in FL systems due to privacy concerns. **We consider the general case where joint distribution p(x,y) for daytime and nighttime can change for real-world FL applications.**
>
> **In Figure 11**, to provide some insights of the data distribution, we plotted the (aggregated) out of vocabulary rates (oov rates) and the number of tokens for a real-world FL application. These metrics show that the **daytime clients have higher oov rates and shorter sentence lengths, and are potentially more difficult to learn.**
>
> **For our simulations, we create different p(x, y) for the daytime and nighttime groups by either changing p(y) or p(x).** We have listed the statistics for the daytime and nighttime subsets in Table 1 in our paper. For EMNIST and CIFAR, the two modes are distinguished by their $p(y)$. For EMNIST, the daytime group has images of digits, while the nighttime group has non-digit characters. For CIFAR, the daytime group has the CIFAR10 data, while the nighttime group has CIFAR100 data, and the two datasets do not have overlapping labels. For Stack Overflow, the two modes are distinguished by their $p(x)$. The daytime mode only has the question sentences, while the nighttime mode only has the answer sentences.
>
> ### 5. Pick the listed related works as baseline methods for comparisons
>
> We thank the reviewer for bringing these works to our attention. We have added detailed discussions of all these works into our related works section, and added comparisons with Semi-cyclic SGD [Eichner et al. 2019], FedEM [Marfoq et al., 2021] and T-shot K-means [Dennis et al. 2021] in our experiments, shown in Figure 2.
>
> **It takes non-trivial modifications for all the listed methods to work in our setting. All these works fail to satisfy at least one of the three properties required by our setting: 1) No strong control on client selection since the distribution is always shifting, e.g., we cannot assume a group of clients are always available. 2) The model cannot use labels for clustering on clients in the test set, to enable rapid deployment to unseen clients. 3) No stateful clients, since in cross-device training, each client usually only participates in a small fraction of rounds. The states can quickly become stale.** Among these papers, [Dennis et al. 2021] has the fewest limitations when applied in our setting, so we add it as a baseline in the experiments. Still, **in order to collect the cluster centers for clients from both modes, we have to enhance the “one-shot” k-means [Dennis et al. 2021] into a “T-shot” clustering**, which collects cluster centers from all clients within the T-round period. There is also no guarantee that clustering on the raw data can be effective when the data distribution in the input space is complicated. Before training, we apply the T-shot clustering on each client’s data, and then use the obtained cluster centers to select the branch for training or inference for each sample.
>
> In addition, we also compare with FedEM [Marfoq et al., 2021], which is recently accepted to NeurIPS. FedEM  requires labeled data to infer the priors and select branches for unseen clients, so we have to let it “cheat” and use all labeled samples for unseen clients to select the branches. Also, unlike our method which only maintains a global Gaussian mixture model, FedEM requires stateful clients that maintain different priors $\pi$ on them, which ideally requires the clients to participate in training in every round. However, for practical cross-device FL and our simulations, each client only participates for at most a few rounds, and FedEM’s performance drops significantly due to the staleness of the states (Figure 1 in [Marfoq et al., 2021]).
>
> Semi-cyclic SGD [Eichner et al., 2019] assumes the system knows whether a client is coming from the daytime group or nighttime group, which is not practical in our cross-device setting, where even labels of the test clients are not available. Therefore, we select the models according to their certainty during test time. For training, we train one of the models during the first half of each period, and the other during the other half.
>
> Also note our original min-loss baseline is effectively adapting IFCA [Ghosh et al., 2020] into our setting, where the adaptation is to select the branches using certainty on the test set.
>
> **As shown in Figure 2, our method is significantly better than these methods in face of periodic distribution shift.**
>
> For more discussions of your suggested related works, please kindly refer to **Clustering and mixture models** in Appendix A.5.

---

> > ### Comment · Reviewer_QWqv · 2021-11-30
> > **Thanks for the response.**
> >
> > Thanks for the authors' comprehensive response.
> >
> > Due to the lack of a real-world dataset with day/night information, I am not convinced that the paper is a proper solution to solve the problem for Diurnal or Nocturnal. Maybe the authors could add some related work to introduce the application problem of Diurnal or Nocturnal in traditional centralised machine learning scenarios?  However, I am ok with the new FL setting for periodically distribution shift with a smooth transition. Therefore, I could increase the score to 5 for the paper's technique contribution.

---

> > > ### Author Response · Authors · 2021-11-30
> > > **Thanks for the suggestion**
> > >
> > > Thank you for the suggestion of citing existing works in centralized scenarios. However, this distribution shift is caused by the different data distributions from users at different time zones (Eichner et al., 2019; Yang et al. 2018), which is unique to online cross-device FL. For online cross-device FL, at different times of the day, the distribution of the participating users' time zones are different, and this changes periodically from day to day. In centralized setting, since it collects the data, even if the data is an online stream with periodically shifting distributions, the server can easily avoid this issue by caching the data, and we are not aware of any related works in this direction. Note that FL is designed to avoid collecting user data.
> > >
> > > We have already showed with Figure 1 (left, middle) that our simulation captures the impact of periodically shifting distribution on training in practice, and in Figure 11 we further give other metrics in a real FL system, indicating the distribution does change smoothly and periodically. Our simulation is more practical than the previous Semi-cyclic SGD setting (Eichner et al., 2019; Ding et al., 2020). We do not claim we have the final solution but we are taking a step towards better simulations and a technique with significant improvements.

---

### Official Review · Reviewer_YzWj · 2021-11-03

**Correctness:** 4
**Technical Novelty And Significance:** 3
**Empirical Novelty And Significance:** 3
**Recommendation:** 6
**Confidence:** 4

**Main Review:**

Strengths:
- The paper proposes a flexible, intuitive and practical algorithm for being able to model distributional shift and incorporate the temporal information through a mixture distribution.
- The multi-branch based network ensures feature parity across different distributions being considered in this paper.
- While the paper is centered around two different distributions, it can be easily extended to multiple distributions to finer granularities depending on the need.
- Experimental results across all the vision and language tasks show an impressive improvement performance.

Cons:
- It seems the authors make an implicit assumption that a client at any given time consists of data from either of the distributions at any given time. Also, at any given time how does one ensure clients are selected from only one of the distributions being considered? As with FL, one can't have any information about the distribution of data on one client.
- The choice of the smooth transition function seems a little ad-hoc. What is the rationale behind selecting such functions? What other functions were considered?
- The performance improvement comes at the cost of added communication complexity and extra hyper parameters to tune.
- How much does the mis-specification of GMM affect the performance of FedTERM. Especially while dealing with clients with fewer samples, the updates for GMM can be high variance prone. How is the performance of FedTERM affected in such scenarios? Please consider adding additional experiments to consider aforementioned scenarios. Finally, misspecification of the prior at each step can lead to the algorithm underperforming severely.

**Summary Of The Paper:**

This paper handles the distributional shift observed in data in clients in a federated learning production setting. In particular, the distributional shift is modeled as a mixture of distributions.Furthermore, a multi-branch network is used to encapsulate the shifting distribution and a Federated Expectation-Maximization algorithm enhanced by Temporal priors of the shifting distribution (FedTEM) is proposed. Experiments for image classification on EMNIST and CIFAR datasets, and next word prediction on the Stack Overflow dataset show the efficacy of the proposed algorithm.

**Summary Of The Review:**

This paper handles the distributional shift observed in data in clients in a federated learning production setting. In particular, the distributional shift is modeled as a mixture of distributions. The innovations include learning a GMM in a federated fashion and the development of the algorithm FedTerm. Experiments show the performance gain with the proposed algorithm. However, there are some gaps in the paper which require additional clarification and experiments. Upon addition of the clarifications and experiments, the paper will be a strong paper to be accepted.

---

> ### Author Response · Authors · 2021-11-17
> **Response to Reviewer YzWj**
>
> Thank you for acknowledging the flexibility and effectiveness of our method. We focus on addressing your concerns in the following.
>
> ### 1. “...at any given time how does one ensure clients are selected from only one of the distributions being considered? As with FL, one can't have any information about the distribution of data on one client.”
> You are right, it is difficult to collect information about the data distribution of individual clients in practical FL systems due to privacy concerns. Nevertheless, **since our goal is to improve federated learning under distribution shifts, we can study the impact of distribution shift on federated training, and compare our simulations with the observations on real data.** As shown in Figure 1, training curves from our simulations match the observations in practical FL systems.
>
> In our simulations, we sample the clients from the predefined daytime and nighttime groups according to Equation 2 and 3. Note that **in our simulations, the distributions of the two groups may have overlapping supports**, e.g., for CIFAR, the “truck” class from the daytime group (CIFAR10) shares some similarities with the “pickup truck” and “tractor” from the nighttime group (CIFAR100); for Stack Overflow, the difference between daytime group (question sentences) and nighttime group (answer sentences) is more subtle and may have more overlaps. Therefore, **the data distribution on the clients from different groups can still share some similarities**.
>
> This overlap assumption is in fact another practical improvement over semi-cyclic SGD [Eichner et al. 2019], which considers worst-case independent distributions. The shared weights in multi-branch networks are designed to capture the overlap of distributions.
>
> ### 2. “What is the rationale behind selecting such functions (to simulate smooth transition)? What other functions were considered?”
>
> We assume the probability of a client to come from one of the modes changes smoothly, so we first considered linear functions, since it is the simplest smooth interpolation. However, the distribution shifts periodically, and periodical linear functions are not smooth at the peaks and valleys, so we also considered the smoother cosine function. **In addition, we considered different $p$’s to control the bias of the distribution towards one mode, to simulate the difference in training speeds between the daytime and nighttime clients (more rounds are finished on the faster group), as observed in (Yang et al., 2019)**. We believe these two function types, together with the different $p$’s, have captured the most important factors of the distribution shift in practice.
>
> ### 3. “The performance improvement comes at the cost of added communication complexity and extra hyper parameters to tune.”
>
> The communication overhead of the additional branch is negligible, since we only pick one linear layer as the branch. **For the models on EMNIST, CIFAR and Stack Overflow, the extra communication overheads are 0.7%, 0.5% and 1.6%, respectively**. We use the same set of hyperparameters  including the prior function (linear or cosine with $p=1$) and label smoothing for different types and $p$’s for the distribution shifts on the same task.
>
> ### 4. “How much does the mis-specification of GMM affect the performance of FedTERM, especially while dealing with clients with fewer samples, the updates for GMM can be high variance prone. How is the performance of FedTERM affected in such scenarios?”
>
> Thanks for the great suggestion. **We provide discussions and results on EMNIST in Appendix A.7 and Figure 10**, where we change the maximum number of training samples on each client, and compare the change in performance for the baselines and FedTEM. In general, fewer training samples can lead to worse test accuracy for all methods, so the accuracy of all three methods degrades as the number of training samples decreases. However, FedTEM maintains its advantages in all cases. We will add results on more tasks later.

---

> > ### Author Response · Authors · 2021-11-22
> > **Updates on the results**
> >
> > We have added the results on CIFAR and Stack Overflow into Figure 10. Our method performs favorably against the baselines under various numbers of training samples per client. We hope you could check these new results and look forward to your further comments, thank you!

---

> > > ### Comment · Reviewer_YzWj · 2021-12-08
> > > **Thanks for the response!**
> > >
> > > Thanks for the comprehensive response! While the experimental results and the method proposed in this paper are solid, the distribution shift being considered is a little constraining. I can see how it is relevant to a production use case. But, in terms of analysis the implicit dichotomy of clients belonging to one distribution at a given time, is a bit limiting.
> > >
> > > Having said that, I would like to maintain my score owing to the flexibility and practical nature of the proposed algorithm.

---

### Official Review · Reviewer_4Moy · 2021-11-03

**Correctness:** 3
**Technical Novelty And Significance:** 2
**Empirical Novelty And Significance:** 2
**Recommendation:** 3
**Confidence:** 4

**Main Review:**

Detailed comments:

(1) The contribution is incremental and lacks sufficient novelty. The multi-branch idea has been fully explored in both multi-task FL and heterogenous/personalized FL. The idea of client clustering has also been studied by several recent works in FL. The proposed solution, i.e., the EM algorithm, is also common for client clustering. These works should be discussed and compared in the paper. Label-smoothing seems helpful here but is also not new.

(2) The newly proposed heuristics, e.g., temporal prior, different shifting schedules (cosine, linear, soft), one-hot cluster assignment, etc., lack rigorous justification. They are too hand-waving. It is not clear why they can be directly integrated into the EM algorithm without hurting its convergence or causing any inconsistency. Moreover, the performance seems to be sensitive to the application of these heuristics.

(3) The datasets/benchmark evaluated in the paper do not have real periodic shifting distributions. The periodic shifting is artificially added but it is not clear why the realistic shifting in practice should follow one of the three schedules and p.

(4) In experiments, when generating the dataset with periodic shifting distributions, q(t) is used to define the periods. However, the algorithm cannot have access to the ground truth q(t) defining the shifting schedule in practice: the application of $\tilde q(t)$ looks like cheating.

(5) There exist many FL methods specifically designed for addressing non-IID distributions across clients and better-personalized models for different clients. However, none of them has been compared in the experiments. "min-loss" is a baseline modified by this paper. Hence, it is hard to justify the advantage of the proposed method.

(6) Fig. 1 is hard to interpret: why do the curves for models trained only during daytime and the ones trained only during nighttime reach peak accuracy at the same time?

(7) The assumption that day-time (night-time) data representation follows a single-mode isotropic Gaussian distribution is too strong and may not hold in practical data.

**Summary Of The Paper:**

his paper proposes a federated learning method to address a specific non-IID challenge over clients, i.e., when some clients are only available during the daytime and others are only available at the night, and these two types of clients' data are drawn from a mixture of two corresponding distributions. They study a multi-branch model, similar to multi-task learning, such that all clients share the same backbone network to extract data representations but a different branch is applied on top of the backbone network for different distributions. Only the clients with the same data distribution use the same branch to produce the final prediction. Under the assumption that the data representation produced by the backbone network follows a mixture of isotropic Gaussian distributions, this paper proposes an EM algorithm to assign clients to different clusters: the E-step determines the (soft) membership of each client belonging to each cluster and assigns it to the branches, while the M-step estimates the mean and variance of the representation for each Gaussian component in the mixture. Some additional heuristics are shown to be important for this method to achieve promising performance, e.g., label smoothing, a temporal prior of the ratio of clients from day and night, linear/cosine/soft schedule of this ratio, one-hot (hard) assignment of clients to clusters, etc. In experiments, they simulate the day-night distribution shift on EMNIST, CIFAR, and Stackflow datasets in FL by splitting each dataset into two parts with different data types. They evaluate the proposed method under different shifting settings (different p) with a few baselines proposed in this paper. They also empirically studied the effects of label smoothing and different period lengths.

**Summary Of The Review:**

The problem of FL with periodically shifting distributions is well-motivated by practical needs. The idea of tackling the non-IID problem in this case by assigning different clients to different branches also makes sense. However, this paper then reduces the problem to FL with client clustering, which is a crowded topic with several similar methods published recently. Given these works, the EM algorithm in this paper is not novel. Moreover, there are a great number of works in the FL community studying the statistical heterogeneity problem and they are all likely to be directly applied here since the day-night shifting is a special case of it. However, the experimental comparison ignores all of them and only compares to baselines developed(modified) in this paper, which fails to provide convincing evidence of the advantage of the proposed method over existing heterogeneous/personalized FL methods. No real day-night periodic shifting exists in any of the three datasets. Knowing the shifting parameter p is somehow cheating. The proposed heuristics are not sufficiently justified whether they are compatible with the EM algorithm and the Gaussian mixture model. Hence, I think there are currently many non-trivial problems that need to be fixed for this paper.

---

> ### Author Response · Authors · 2021-11-17
> **Response to Reviewer 4Moy (1)**
>
> Thank you for acknowledging the importance of studying periodically shifting distributions in federated learning, which exists in practical cross-device FL systems, but has not been well studied in previous works for client clustering and personalization. **We notice some misunderstandings of the key points of our method in the comments, and hope our responses could address your concerns.** We have edited the draft to make these points clearer.
>
> ### 1. Misunderstanding: *”(4) Knowing the shifting parameter p is somehow cheating”*
>
> **We do not assume the shift parameter $p$ is known to our algorithm.** In our updated version, we use $q_{C, p}$ and $q_{L, p}$ to represent the cosine and linear smooth transition function to simulate periodical distribution shift, and $\tilde{q}=q_{C, 1}$ and $\tilde{q}=q_{L, 1}$ to represent the linear and cosine temporal priors used in our algorithm. Though we introduce a hyperparameter $p$ to simulate various periodical distribution shifts in practice, the temporal prior used in our algorithm has a fixed $p=1$, which is used throughout the experiments. Moreover, linear prior $\tilde{q}=q_{L, 1}$ can achieve better results on vision tasks (EMNIST and CIFAR) and cosine prior $\tilde{q}=q_{C, 1}$ can achieve better results on the language task (Stack Overflow), regardless of the smooth transition function for data simulation. We also evaluated a weaker temporal prior without assuming the shape of the function, which can improve over other baselines despite being inferior to $\tilde{q}=q_{C, 1}$ and $\tilde{q}=q_{L, 1}$.
>
> ### 2. *“(1) The contribution is incremental and lacks sufficient novelty”*. Could you elaborate on  which works *“should be discussed and compared in the paper”*?
>
> **We discussed multi-task FL and heterogenous/personalized FL in our introduction and related works sections. Per reviewer’s request, we have updated the related works in Appendix A.5, and added experiments to compare with more methods in Figure 2.**  Since the reviewer did not provide enough details on which works we missed , we do our best to highlight the challenges of applying previous works to tackle the periodical distribution shift . **If these cannot address the reviewer’s concerns, we sincerely hope the reviewer can be more specific on which works to compare with during the discussion phase.**
>
> Note that the baselines we added in Figure 2 are often variants of the previous methods, as they cannot be directly applied to the practical cross-device FL setting for reasons listed below.
> - [Smith et al.; 2017; Marfoq et al., 2021; Briggs et al., 2020; Fu et al., 2021; Xie et al., 2021; Andreux et al., 2020] assume clients will participate every round, and each client will maintain a local state, which is impractical for large-scale cross device FL systems.
> - [Ghosh et al., 2020; Mansour et al., 2020; Marfoq et al., 2021, Briggs et al., 2020; Fu et al., 2021; Duan et al., 2021; Xie et al., 2021] assume each test client has labeled training data for personalization.
>
> To our knowledge, only Semi-cyclic SGD [Eichner et al., 2019] and [Ding et al., 2020] consider periodical distribution shifts. Please kindly refer to **Methods to compare** in Section 4 for our adaptations to methods in Figure 2. For more detailed comparisons with existing works, please kindly refer to Appendix A.5 of the current version.

---

> ### Author Response · Authors · 2021-11-17
> **Response to Reviewer 4Moy (2)**
>
> ### 3. *“(2) It is not clear why they can be directly integrated into the EM algorithm without hurting its convergence or causing any inconsistency…The proposed heuristics are not sufficiently justified whether they are compatible with the EM algorithm and the Gaussian mixture model. Moreover, the performance seems to be sensitive to the application of these heuristics.”*
>
> **The empirical convergence of the proposed FedTEM algorithm is verified by extensive experiments with superior performance and ablation studies for the priors and regularizations.  We also highlight the difficulty of theoretical convergence guarantees for a stochastic EM algorithm below.**
>
> If we assume all clients will participate every round (removing the stochasticity), and add other unrealistic assumptions (see Remark 1 of FedEM [Marfoq et al., 2021]),  we can get similar convergence guarantees for the proposed FedTEM. However, the periodical distribution shift problem itself will not happen if all clients will participate in every round.
>
> Another challenge is the heterogeneity in federated learning. Even if all clients can participate every round, the EM algorithm can only converge to a local solution. There are potentially many local solutions due to the heterogeneity and the temporal prior is intuitively beneficial to guide the training process to a preferred solution. We acknowledge that new theoretical frameworks are desirable, but that is beyond the scope of this paper.
>
> We show that under periodical distribution shifts, our algorithm outperforms many other methods including FedEM, and can even outperform models trained without distribution shift. See updated Figure 2.
>
> **Could the reviewer elaborate on “the performance seems to be sensitive to the application of these heuristics”?** Through extensive ablation study, one set of hyperparameters (including those for label smoothing and the prior) transfer well across different types and $p$’s of the distribution shift for a specific task.  See Figure 3 and 4.
>
> ### 4. "*(3) The datasets/benchmark evaluated in the paper do not have real periodic shifting distributions. The periodic shifting is artificially added but it is not clear why the realistic shifting in practice should follow one of the three schedules and p.*"
>
> **We acknowledge the importance of real datasets with natural distribution shifts. However, we highlight the difficulty of collecting a new dataset and believe that is beyond the scope of this draft. The importance of creating artificial datasets that align well with practical observations should not be underestimated.** The seminal FedAvg paper [Mcmahan et al. 2017] used partitioned MNIST and CIFAR for experiments. Many seminal works in other disciplines, such as continual learning [1] and meta learning [2], also modified  existing datasets to simulate streaming setting or unseen new tasks.
>
> The reviewer mentioned “three schedules” in the comment. To clarify, when simulating distribution shift, we only consider **two schedules (linear and cosine)**. $p$ is used to simulate the difference in training speed between daytime and nighttime clients in practice [3], since it controls the bias of the distribution towards one mode in our simulations.
>
> **It is natural to assume the distribution changes smoothly in practice. Linear interpolation is the simplest smooth interpolation, and cosine functions are more smooth than periodical linear functions. We have verified in Figure 1 that our simulation behaves similarly to practical FL training on real data, and it is a more practical simulation than the Semi-cyclic SGD assumption [Eichner et al. 2019].**
>
> ### 5. Misunderstanding and our updates about *“(5) However, none of them (existing FL methods addressing non-IID or personalization) has been compared in the experiments. "min-loss" is a baseline modified by this paper.”*
>
> **Min-loss is not invented by us; it is a variant of  IFCA [Ghosh et al. 2020].** IFCA assumed the existence of labeled data on testing clients and did not consider unseen clients with no labeled data as in our setting. We use certainty instead of the loss (needs labels to compute) to select branches during inference as a variant. We also find the label smoothing regularization to benefit IFCA in the multi-branch network, so we add this regularization for fair comparisons. **In addition, we have added comparisons with Semi-cyclic SGD, FedEM and T-shot K-means in Figure 2 of our latest version.** As mentioned before, **it takes non-trivial changes for almost all previous methods to work in our setting**.  Please kindly refer to **Methods to compare** in Section 4 for how we adapted these methods into our setting.

---

> ### Author Response · Authors · 2021-11-17
> **Response to Reviewer 4Moy (3)**
>
> ### 6. *“(6) Fig. 1 is hard to interpret: why do the curves for models trained only during daytime and the ones trained only during nighttime reach peak accuracy at the same time?”*
>
> **These models reach peak accuracy simultaneously when the data distribution is more biased towards the simpler mode.** Data from two modes usually have different difficulties for the model to generalize. We have plotted the out of vocabulary rates (oov rates) and the number of tokens at each round in the real FL system in Figure 11. These metrics indicate that the daytime clients have higher oov rates and shorter sentence lengths, indicating these sentences might be more different to learn. This has also been observed in Figure 4 of [3]. **Note this surprising observation exists for both real data and our simulations, further justifying our simulation captures important characteristics of the real data.**
>
> ### 7. “(7) The assumption that day-time (night-time) data representation follows a single-mode isotropic Gaussian distribution is too strong and may not hold in practical data.”
>
>
> Our model assumes each mode of data can be represented by a multivariate Gaussian distribution in the feature space. Since we jointly train the mixture model with the network, this can be achieved by utilizing the expressive power of neural networks and let it learn to map the data into such a feature space. Similar assumptions have also been adopted in many successful works on real data, such as VAEs. So this is not a strong assumption.
>
> ### Additional References
>
> [1] Kirkpatrick, James, Razvan Pascanu, Neil Rabinowitz, Joel Veness, Guillaume Desjardins, Andrei A. Rusu, Kieran Milan et al. "Overcoming catastrophic forgetting in neural networks." Proceedings of the national academy of sciences (2017).
>
> [2] Finn, Chelsea, Pieter Abbeel, and Sergey Levine. "Model-agnostic meta-learning for fast adaptation of deep networks." In ICML 2017.
>
> [3] Yang, Timothy, Galen Andrew, Hubert Eichner, Haicheng Sun, Wei Li, Nicholas Kong, Daniel Ramage, and Françoise Beaufays. "Applied federated learning: Improving google keyboard query suggestions." arXiv:1812.02903.

---

### Author Response · Authors · 2021-11-17
**Summary of updates**

We thank all the reviewers for their valuable comments. We have updated our paper accordingly. We summarize the main updates as below:

1. Per reviewers’ requests, we have added comparisons with Semi-cyclic SGD [Eichner et al., 2019], One-shot K-means [Dennis et al., 2021], and FedEM [Marfoq et al., 2021] in the experiments (Figure 2). Together with IFCA (Min Loss) and two FedAvg baselines (with or without distribution shift), we have compared with 6 baselines. Note that except for FedAvg, we need non-trivial modifications to other methods to make them applicable in our setting. Please refer to the experiment section for more details.

2. We updated the related works in Appendix A.5 to include detailed discussions with previous methods for clustered federated learning and personalization, highlighting the key challenges of our setting and potential limitations of existing approaches. **We emphasize that most of the previous works cannot be directly applied in our setting without non-trivial modifications.** Semi-cyclic SGD [Eichner et al., 2019] and [Ding et al., 2020] are the only previous works we are aware of that explicitly consider periodical distribution shift in FL.

3. Comparing the effect of using multi-branch network instead of single-branch network for the "Vanilla” and "No Dist. Shift” baselines in Figure 9 of Appendix A.6.

4. Quantifying the effect of using fewer training samples per client on our EM algorithm, shown in Figure 10 of Appendix A.7.

5. Adding the out of vocabulary rates and number of tokens during training a language model in a real FL system, shown in Figure 11 and Appendix A.8.

6. Changing the plots into color blind friendly.

Please let us know if you have any further questions, thank you!

---

### Author Response · Authors · 2021-11-29
**Looking forward to your comments!**

The discussion period is ending today but we have not received any further feedback from the reviewers. We appreciate the opportunity to have further discussions with you and incorporate your suggestions into the next version of our paper.

We hope our responses have addressed the concerns of the reviewers, especially for Reviewer 4Moy and QWqv. For Reviewer 4Moy, **our method does not assume knowing the exact distribution shift.** As suggested by Reviewer QWqv, we have also added more comparisons with existing works in clustered federated learning and personalization, in both the experiments and related works, though non-trivial changes are needed  for these methods to work in the practical cross-device setting, further demonstrating the practical importance of our method.

---

### Decision · Program_Chairs · 2022-01-20

**Decision:**

Accept (Poster)

**Comment:**

The paper considers FL with periodically shifting distributions, which is a very relevant and timely research question in the area of federated learning, and learning under distributions shifts. The paper proposed an interesting unsupervised way to learn grouping clients into different branches during training, using a federated version of the EM algorithm. Overall the paper contains several solid contributions in some novel combinations, but remained borderline in terms over overall scores. While reviewers were generally positive about the approach, technical soundness and the importance of the question, still some concerns remained.

Concerns included on the level of novelty relative to several recent similar related FL works, and them being included as baselines. Several of these are now discussed in the rebuttal and revisions, but not all in sufficient depth. While the datasets used seem to offer sufficiently hard task from the split between the 'day' and 'night' distribution, several questions were raised if the treatment of priors is realistic enough. This includes the question of fair hyperparameter tuning with respect to the temporal priors, as well as potential misspecification of the same, towards a more principled treatment of the priors. The authors have answered several of the concerns in the revision, and have added more baseline comparisons, raising the paper narrowly above the acceptance bar in my assessment.

Time-wise, the very related paper Marfoq et al 2021 "Federated Multi-Task Learning under a Mixture of Distributions" seems to have been available 6 weeks before ICLR deadline. We thank the authors for having included it in discussion and experiments, but the discussion of related contributions needs to be expanded (main difference seems to be supervised vs unsupervised group assignment).

My impression is these points can be addressed in a camera ready version, and I hope the detailed feedback here by all reviewers below will be incorporated.